# LCEN: A Nonlinear, Interpretable Feature Selection and Machine Learning Algorithm

**Pedro Seber**                                                                           *pseber@mit.edu*
*Department of Chemical Engineering*
*Massachusetts Institute of Technology*

**Richard D. Braatz**                                                                     *braatz@mit.edu*
*Department of Chemical Engineering*
*Massachusetts Institute of Technology*

**Reviewed on OpenReview:** *https://openreview.net/forum?id=wmNucISPdl*

## Abstract

Interpretable models can have advantages over black-box models, and interpretability is essential for the application of machine learning in critical settings, such as aviation or medicine. This article introduces the LASSO-Clip-EN (LCEN) algorithm for nonlinear, interpretable feature selection and machine learning modeling. In a wide variety of artificial and empirical datasets, LCEN constructed sparse and frequently more accurate models than other methods, including sparse, nonlinear methods, on tested datasets. LCEN was empirically observed to be robust against many issues typically present in datasets and modeling, including noise, multicollinearity, and data scarcity. As a feature selection algorithm, LCEN matched or surpassed the thresholded elastic net but was, on average, 10.3-fold faster based on our experiments. LCEN for feature selection can also rediscover multiple physical laws from empirical data. As a machine learning algorithm, when tested on processes with no known physical laws, LCEN achieved better results than many other dense and sparse methods — including being comparable to or better than ANNs on multiple datasets.

## 1 Introduction

Many modeling methods and algorithms exist, including for the construction of linear, ensemble-based, and deep learning models. Complex models can have greater capability to model phenomena due to their lower bias, but have intricate and numerous mathematical transformations that prevent humans from understanding how an output was predicted by a model, or the relative or absolute importance of the inputs. Moreover, a lack of transparency may prevent the model from being trusted in critical or sensitive applications (Hong et al., 2020).

In a modeling context, interpretability can be defined as "how an output $y = f(X)$ was predicted for a given input $X$ — that is, provide $f(\cdot)$ in a form readily understandable to humans," which also allows the model's outputs to be explainable. There are two main methods to increase interpretability: the use of model-agnostic algorithms, which extract interpretable explanations *a posteriori* and work for any model, or the direct use of interpretable models (Ribeiro et al., 2016). Interpretable models include "decision trees, rules, additive models, attention-based networks, and sparse linear models" (Ribeiro et al., 2016).[1] Interpretable models can have many advantages over black-box or *a posteriori* explanations, including the ability to assist researchers in refining the model and data, or better highlighting scenarios in which the model fails or lacks robustness (Rudin, 2019). Special attention should be given to sparse models, which identify the most important features, can make the model more robust to variations in the input data, and can significantly improve the model's interpretability if an interpretable model is used (Rudin, 2019). A sparse model may

---

[1]Nonlinear models may also be made sparse, and even interpretable, as described in this and many works.

be defined as "a model that uses few input features, particularly relative to the total number of features available." However, even a linear model or decision tree/rules can become unwieldy and challenging to interpret if hundreds or thousands of coefficients or rules are present. A recent work states that sparsity may be achieved for individual predictions even if the overall model is not sparse (Sun et al., 2024).

Feature selection is the process of selecting the most important features in a model to increase its robustness, interpretability, or sparsity. Many criteria for feature selection exist (Heinze et al., 2018), including significance based on p-values (using a univariate, iterative/stepwise, or global method), using information criteria (such as the AIC (Akaike, 1974) and BIC (Schwarz, 1978)), using penalties (such as in LASSO (Santosa & Symes, 1986; Tibshirani, 1996) and elastic net (EN) (Zou & Hastie, 2005)), criteria based on changes in estimates, and expert knowledge. More broadly, these methods can be classified as filter, wrapper, or integrated methods. While no method is superior for all problems, different works have evaluated and criticized these criteria. For example, stepwise regression is one of the most commonly used methods in many fields thanks to its computational simplicity and ease of understanding (Whittingham et al., 2006; Heinze et al., 2018; Smith, 2018). However, stepwise regression is prone to ignoring features with causal effects, including irrelevant features, generating excessively small confidence intervals, and producing incorrect/biased parameters (Whittingham et al., 2006; Smith, 2018). LASSO is simple and computationally cheap, and has performed well for some problems (Hebiri & Lederer, 2013; Tian et al., 2015; Pavlou et al., 2016), but can overselect irrelevant variables, tends to select only as many features as there are samples, and does not handle multicollinear data well (Heinze et al., 2018; Zou & Hastie, 2005).[2]

Originally, most feature selection methods applied only in linear contexts (or were applied primarily in linear contexts). Highlighting this, the only sparse models referenced in the highly cited review by Ribeiro et al. (2016) are linear. The most commonly used sparse methods (LASSO, EN, and their variants) are linear regressors. To address this limitation, later works consider sparse nonlinear models. For example, McConaghy (2011), Brewick et al. (2017), and Sun & Braatz (2020) defined sets of features consisting of polynomials (all works), interactions (all works), and/or non-polynomials (McConaghy, 2011; Sun & Braatz, 2020). ALVEN (Sun & Braatz, 2020) uses an F-test for each feature (including the expanded set of features) to determine whether to keep a feature in the final EN model, a filter approach. However, this F-test has very poor feature selectivity, as nearly all features are selected when traditional values of $\alpha$ ($0.001 \leq \alpha \leq 0.05$) are used. Furthermore, the ordering of the features with respect to their p-values does not follow their relevance, as many irrelevant features are among those with the lowest p-values, and relevant features can be among those with the highest p-values (including $p \gg 0.05$). Other relevant methods include the smoothly clipped absolute deviation (SCAD) (Fan & Li, 2001); Sparse Additive Models (Ravikumar et al., 2009), which were used with $L_0$ and $L_2$ regularization by Liu et al. (2022); the adaptive elastic net (Zou & Zhang, 2009); the minimax concave penalty (MCP/MC+) (Zhang, 2010); the thresholded LASSO (van de Geer et al., 2011); the two-stage regularized method of De Mol et al. (2009); two forms of modified, nonlinear LASSO algorithms (Yamada et al., 2018); a cutting plane algorithm (Bertsimas & Parys, 2020); Bayesian symbolic regression approaches (Xu et al., 2021); and the UniLasso (Chatterjee et al., 2025).

More recently, $L_1$ regularization has been applied to neural networks for nonlinear feature selection. In its simplest form, group LASSO is applied to zero all the outputs of some neurons (sparsifying the network and eliminating features when zeroing input-layer neurons) (Dinh & Ho, 2020; Scardapane et al., 2017; Wang et al., 2021). LassoNet, a slightly modified version of this algorithm, applies the $L_1$ penalty only to the input layer and includes a skip-connection between that layer and the output layer (Lemhadri et al., 2021). More complex applications of this method include the multi-modal neural networks of Zhao et al. (2015), the concrete autoencoders of Balın et al. (2019), and the teacher-student network of Mirzaei et al. (2020). The first two methods are at least partially unsupervised, suggesting that they can select the most relevant features for a given dataset irrespective of the task. Some works have used approaches other than the $L_1$ norm for neural network-based feature selection, such as the $L_0$ pseudonorm (Yamada et al., 2020). While these deep learning models are powerful tools, two considerable limitations are present: first, they do not provide any information on how the selected features are contributing to the final prediction, significantly limiting interpretability. Although sometimes useful, *a posteriori* methods to extract this information have

---

[2]This last point is somewhat controversial in the literature; see Hebiri & Lederer (2013) and Dalalyan et al. (2017), for example.

been found unreliable (Rudin, 2019). Second, these complex model architectures may take "shortcuts" to make apparently accurate predictions (Lapuschkin et al., 2019; Rosenzweig et al., 2021). However, these "shortcuts" are not really relevant to the task, preventing proper generalization and human interpretation.

To create nonlinear, interpretable, and sparse machine learning models with high predictive and descriptive power, we propose the LASSO-Clip-EN (LCEN) algorithm. This algorithm generates an expanded set of nonlinear features (such as in ALVEN) and performs feature selection and model fitting. This feature set expansion, together with the Clip step, provide LCEN with the ability to do nonlinear predictions. The feature selection algorithm and the specific usage of thresholded LASSO (LC) followed by thresholded EN (ENC) in a combined algorithm are the basis of novelty in this work. The algorithmic structure of LCEN is motivated by past works that proved desirable theoretical properties of the thresholded LASSO (a LASSO-Clip model) and the thresholded EN (an EN-Clip model), which include Zhou (2009), Meinshausen & Yu (2009), Zou & Zhang (2009), Zhou (2010), and van de Geer et al. (2011). Variations and ablations of LCEN are observed to not perform as well as LCEN. In case studies involving artificial and empirical data, LCEN successfully rediscovers physical laws from data belonging to multiple different areas of knowledge with errors $< 0.5\%$ on the coefficients, a value within the empirical noise of the datasets. For tested datasets from processes whose underlying physical laws are not yet known, LCEN is observed to attain lower root-mean-square errors (RMSEs) than many sparse and dense methods, leads to sparser models than all but two methods tested,[3] and simultaneously ran faster than most alternative methods.

## 2 Methods

### 2.1 The LCEN algorithm

The LCEN algorithm (Algorithm 1) begins with the LASSO step, which temporarily sets the *l1_ratio* to 1. Five-fold cross-validation (CV) on the training set is employed among all combinations of *alpha*, *degree*, and *lag* values. The training dataset is split randomly for each fold (as per sklearn's KFold function). First, additional features are temporarily appended to the data based on the *degree*, *lag*, *trans_type*, *interaction*, and *transform_y* hyperparameters. We developed a custom algorithm (Algorithm 2) to perform this feature expansion. Due to this dependency on the *degree* and *lag* hyperparameters, this feature expansion occurs within the cross-validation process, creating a temporary augmented dataset that is scaled to have mean = 0 and standard deviation = 1 and then input to the LASSO method. For each hyperparameter combination and fold, a validation mean-squared error (MSE) is recorded. The values of *degree* and *lag* corresponding to the LASSO model with the lowest validation MSE (averaged across all five folds) are recorded, and a LASSO model using this combination of hyperparameters is fit using the training data to obtain scaled parameters (estimated coefficients).

The next step in the LCEN algorithm is the clip (thresholding) step, in which the features whose scaled LASSO parameters have absolute values smaller than the *cutoff* hyperparameter are recorded so that they can be removed from the expanded dataset, and their coefficients are forced to 0. This step reduces the number of features to be considered, speeding up the algorithm and increasing the accuracy of the model predictions by removing irrelevant/less relevant features.

The EN step involves cross-validation on the training set among all combinations of *alpha* and *l1_ratio*, using the values of *degree* and *lag* obtained in the LASSO step. Once again, the training dataset is split randomly for each fold (as per sklearn's KFold function) and the features are expanded and scaled, then the features recorded in the Clip step are removed. For each hyperparameter combination and fold, a validation MSE is recorded. The values of *alpha* and *l1_ratio* corresponding to the EN model with the lowest validation MSE (averaged across all five folds) are recorded, and an EN model using this combination of hyperparameters is fit using the training data to obtain new scaled parameters.

A second clip (thresholding) step is applied to these EN-scaled parameters, zeroing the coefficients of the features whose scaled EN parameters have absolute values smaller than the *cutoff* hyperparameter. Lastly, some post-processing steps are performed. The scaled coefficients are unscaled by multiplying the scaled

---

[3]LCEN is approximately equivalent to other LASSO-based methods in terms of sparsity.

coefficients by the standard deviation of the y training data and dividing by the standard deviation of each corresponding X feature. Then, a dot product of the train or test data and the unscaled coefficients is taken to obtain the final predictions. This procedure returns the trained EN model after the second clip step, which is interpretable and nonlinear, and the predictions made with the unscaled coefficients on the data.

---

**Algorithm 1** LASSO-Clip-EN (LCEN)

---

**Input:** X and y data; lists of hyperparameters *alpha*, *l1_ratio*, *degree*, *lag*; hyperparameters *cutoff*, *trans_type*, *interaction*, *transform_y*

# LASSO step: filters features without requiring a combinatorially large number of potential hyperparameters, as *l1_ratio* is fixed. Determines the *degree* and *lag* hyperparameters for feature expansion.

Temporarily set *l1_ratio* = 1.

**for** each hyperparameter combination in (*alpha* × *degree* × *lag*) **do**

    Generate additional features based on the *trans_type*, *interaction*, *transform_y*, the current *degree*, and the current *lag* hyperparameters [Algorithm 2].

    Temporarily append the new features to the X data for cross-validation.

    Scale the data such that each feature's mean = 0 and its standard deviation = 1.

    Perform five-fold cross-validation with a user-selected CV algorithm and the LASSO method.

    For each fold, record the validation MSE for this hyperparameter combination.

**end for**

Obtain the combination of hyperparameters with the lowest average validation MSE from the above cross-validation. Record the best *degree* and *lag* hyperparameters.

Fit a LASSO model on the scaled training data with these hyperparameters to obtain parameters.

# Clip step: further reduces the number of features, speeding up the model and increasing sparsity.

Record all features whose scaled parameters have absolute values < *cutoff* from the training and test data for removal during the EN training.

# EN step: trains the final model on the features that passed the LASSO and Clip steps. EN is used for its machine learning performance and stability.

Restore *l1_ratio* to its original list of values.

**for** each hyperparameter combination in (*alpha* × *l1_ratio*) **do**

    Generate additional features based on the *trans_type*, *interaction*, *transform_y*, the optimal *degree*, and the optimal *lag* hyperparameters [Algorithm 2].

    Temporarily append the new features to the X data for cross-validation.

    Remove the features not selected by LASSO or recorded in the Clip step.

    Scale the data such that each feature's mean = 0 and its standard deviation = 1.

    Perform five-fold cross-validation with a user-selected CV algorithm and the EN method.

    For each fold, record the validation MSE for this hyperparameter combination.

**end for**

Obtain the combination of hyperparameters with the lowest average validation MSE from the above cross-validation. Record the best *alpha* and *l1_ratio* hyperparameters.

Fit an EN model on the scaled training data with these hyperparameters to obtain parameters.

# Clip step II

Remove all features whose scaled parameters have absolute values < *cutoff*.

# Post-processing: returns the model with coefficients, predictions, and metrics in a human-readable way.

Unscale the coefficients of the selected parameters based on the standard deviations of the data.

Obtain train/test predictions by performing a dot product of the unscaled coefficients with the expanded train/test data containing only the selected features.

**return** the trained EN model, the predictions, and the evaluation metrics.

---

The LCEN algorithm (Algorithm 1) has five hyperparameters: *alpha*, which determines the regularization strength (as in the LASSO, EN, and similar algorithms); *l1_ratio*, which determines how much of the regularization of the EN step depends on the 1-norm as opposed to the 2-norm (as in the EN algorithm); *degree*, which determines the maximum degree for the basis expansion of the data (Table A2); *lag*, which determines the maximum number of previous time steps from which $X$ and $y$ features are included (relevant

only for dynamic models); and *cutoff*, which determines the minimum value a scaled parameter needs to have to not be eliminated during the clip steps. Details on the cross-validated hyperparameter values for all models are in Section A3. Three other hyperparameters are relevant to the expansion of features (Algorithm 2) but do not interact with the LCEN algorithm directly. The *trans_type* hyperparameter controls what kinds of features are appended to the data. It can be set to 'all' to include all transforms (see Table A2 for an example of what features are included), 'poly' to include only polynomial and interaction terms, and 'simple_interaction' to include only interaction terms. The *interaction* hyperparameter is a boolean that controls whether interaction terms are included in the feature expansion process. Finally, the *transform_y* hyperparameter, which is relevant only when $lag > 0$, is a boolean that determines whether the y features from previous time steps will also be transformed based on *trans_type* or only the raw values of y from previous time steps will be included.

LCEN scales similarly to the thresholded and regular EN. For an $N \times P$ dataset under $F$-fold cross-validation with $A$ potential $\alpha$ values, $L$ potential $L_1$ ratio values, and $C$ potential *cutoff* values, LCEN scales as $\mathrm{O}(NP^2 FALC)$. The *degree* hyperparameter increases the number of features $P$ in a supralinear way. The *lag* hyperparameter increases the number of features $P$ in a linear way. Conversely, higher values for the *cutoff* hyperparameter decrease the number of features $P$.

## 2.2 LCEN is optimal when compared to ablated and variant algorithms

The rationale behind this sequence of steps — LASSO, then Clip, then EN, then a second Clip — seeks to balance the algorithm's high accuracy and sparsity with a low runtime. As shown by the ablation experiments (Section A4 and Table 2), other combinations/variants do not achieve the same performance as LCEN. Specifically, as feature selection algorithms, the LCEN, LASSO-Clip (LC, the thresholded LASSO), and LASSO-Clip-LASSO (LCL) algorithms perform much more accurate nonlinear feature selection than LASSO-EN (LEN) or the methods starting with EN. Moreover, the methods that start with the LASSO are considerably faster. As machine learning algorithms, the LASSO-Clip, EN-Clip (ENC, the thresholded EN), and LASSO-EN combinations tend to have lower accuracy than LCEN. The EN-Clip combination is also much slower and less sparse than LCEN, and the LASSO-EN combination is slower and slightly less sparse. The LASSO-Clip-LASSO combination is less accurate than LCEN, although it is slightly faster and sparser. The EN-Clip-EN (ENCEN) combination achieves similar accuracy, but is much slower and less sparse than LCEN. It is possible to add a debiasing step at the end by using ordinary least-squares (OLS) to estimate the coefficients of the features selected by LCEN (Belloni & Chernozhukov, 2013). This LCEN→OLS variant model estimates coefficients more accurately for one of the ablation experiments, but performs worse in terms of test-set MSE on empirical datasets.

The combinations that start with EN are slower than LCEN because EN has a greater number of hyperparameter combinations to be tested, and these combinations are tested with a higher number of features (as the full expanded feature set has not been subject to any selection via the LASSO and Clip steps). These combinations are also less sparse because the $L_1$ and $L_2$ norms compete during EN regularization, and a combination that prioritizes the $L_2$ norm may have a lower cross-validation MSE. Beginning with the LASSO increases the algorithm's sparsity and speed at no accuracy cost.

The use of hard-thresholding (Clip) steps improves LCEN's accuracy and sparsity. The increase in sparsity also lowers the algorithm's runtime. The second Clip step is less impactful than the first and does not affect the algorithm's runtime, but it still improves LCEN's feature selection capabilities. We highlight that the Clip steps operate on the scaled parameters; thus, any issues that may arise due to the (relative) magnitude of the coefficients are not significant.

## 2.3 Experimental setup

Multiple datasets (summarized in Table A1) are used to test the performance of the LCEN algorithm. These datasets can be divided into three categories: artificial data, empirical data from processes with known physical laws, and empirical data from processes with no known physical laws. Further description of these datasets and how the artificial datasets are generated is available in Section A2; the empirical datasets are also described in Section 3.2. All models tested in this work had their hyperparameters selected by

5-fold cross-validation (CV) and this CV procedure was repeated for 3 different seeds so that the average ± standard deviation of results can be reported, except for those trained on the "GEFCom 2014" dataset, which used time series cross-validation. For the "Diesel Freezing Point" dataset, an experiment comparing 10-fold CV with 5-fold CV was performed (Table A10). The separation between training and testing sets varied depending on the dataset and is detailed in Section A2.

## 3    Results

### 3.1    In artificial datasets, LCEN provides high feature selection performance and robustness to noise and multicollinearity

LCEN is evaluated both as a feature selection algorithm (this section and Table 2) and as a machine learning algorithm (rest of Section 3.2). The first datasets used to validate the LCEN algorithm are multiple linear datasets ("Artificial linear"). These datasets feature all combinations of {100, 500, 1000} samples × {100, 500, 1000} true features × {0%, 25%} noise level × {25%, 50%, 75%, 100%} additional false features. The noise level is defined as mean(added noise/noiseless y)×100%. The added Gaussian noise has $\mu = 0$ and a suitable $\sigma$ to reach the desired noise level. For each combination, 3 repeats with different random seeds were created. This experiment contains multiple challenging conditions, including cases where the number of features was much larger than the number of samples ($P \gg N$), cases with a significant proportion of false features, and cases with a high noise (low correlation between the input $X$ and the output $y$).

The methods LASSO, EN, fastSparseGAMs (FS-GAMs) (Liu et al., 2022), SCAD, MCP, symbolic regression (SymReg) (implemented by Stephens et al. (2022)), thresholded EN, and LCEN were tested on this feature selection task. Overall, LCEN consistently performed similarly to the thresholded EN and outperformed all other methods in this task, as measured by the Matthews Correlation Coefficient (MCC) (Figs. A1–A6). Furthermore, LCEN was 4.7-fold faster than the thresholded EN on these datasets, and 10.3-fold faster than the thresholded EN on average (Table A3). EN was typically completely unselective, classifying all features as true except in a few scenarios with $N > P$. SymReg performed marginally better, but still had a very low overall performance. LASSO, FS-GAMs, SCAD, and MCP performed better than EN and SymReg, having similar performances among themselves; notable exceptions were cases in which LASSO was completely unselective, and most scenarios with $N > P$ and 0% noise, which allowed SCAD and MCP to perform perfect classification. LCEN performed perfect classification in the scenarios with $N > P$ and 0% noise even more frequently than SCAD and MCP, and surpassed the methods (other than thresholded EN) in the other scenarios in terms of absolute MCC by 19.8% on average when $N = P$ and by 8.2% on average when $N < P$.

The first step of LCEN uses LASSO, which has been stated to underperform with multicollinear data (Heinze et al., 2018; Zou & Hastie, 2005). Therefore, tests using multicollinear data are performed next. The goal is to assess whether LCEN can determine the presence of two different but correlated variables more accurately than LASSO. Noise $\epsilon_1$, at different levels (as defined above), was added to the $X_0$ variable to create a correlated variable $X_1$. A second noise $\epsilon_2$, also at different levels, was added to the final $y$ data. When $\epsilon_1 = 0$, both variables are equal and separation is not possible. However, for other $\epsilon_1$ values, LCEN successfully identifies that two relevant variables exist and assigns correct coefficients to them (Fig. A7). Specifically, when the noise level $\epsilon_1$ associated with the $X$ data (which indicates how different the $X$ variables are, as highlighted by the variance inflation factors [VIFs] in Fig. A7) is greater than the noise level $\epsilon_2$ associated with the $y$ data, LCEN separates both variables with coefficient errors ≤ 5%. When both noise levels are similar, LCEN separates both variables with coefficient errors between 5% and 10%. The $X$ data used in this experiment has very high multicollinearity (as shown by the VIFs); real data will typically have lower VIFs and thus be easier for LCEN to separate.

Next, a more complex equation is used to further validate LCEN. The "Relativistic energy" data contain mass and velocity values used to calculate $E^2 = c^4 m^2 + c^2 m^2 v^2$. As before, datasets with increasing noise levels are created. The *degree* hyperparameter is allowed to vary between 1 and 6 in this experiment. These *degree* values lead to expanded datasets with {8, 22, 42, 68, 100, 138} features respectively. Although there are only two true features, there are a significant number of false features, many of which have similar functional

forms to the true features. LCEN selected only relevant features for all noise levels tested ($\leq 20\%$), and the coefficients were typically equal to the ground truth (Table 1). The only major divergence happened at a noise level of 20%, as the coefficient for $m^2$ had a 25% relative error. This error led to our hypothesizing that it is challenging to distinguish among the features involving $m$ (such as $m$, $m^2$, and $m^3$) due to the low range of the data. Thus, another dataset with the same properties but a larger range of values for $m$ was created. LCEN performed better on this dataset, again selecting only relevant features for all noise levels tested ($\leq 30\%$) and having much lower errors in the estimated coefficients (Table 1). These experiments consistently show the robustness of LCEN and how the range of the data can affect predictions. To clarify our design choices and the relevance of each individual part of the LCEN algorithm, ablation tests are performed with this dataset in Section A4 of the Appendix.

**Table 1:** Coefficient values and corresponding relative error to the ground truth for the "Relativistic energy" dataset at different noise levels. The first coefficient is for $m^2$ and should be $c^4 = 8.078{\times}10^{33}$ m$^4$/s$^4$; the second coefficient is for $m^2v^2$ and should be $c^2 = 8.988{\times}10^{16}$ m$^2$/s$^2$. The left table is for the dataset with $1 \leq m < 10$, and the right table is for the dataset with $1 \leq m < 100$.

| Noise | Coefficients | Error (%) |
|---|---|---|
| 0% | $8.077{\times}10^{33}$, $8.987{\times}10^{16}$ | 0.013, 0.009 |
| 5% | $8.081{\times}10^{33}$, $8.969{\times}10^{16}$ | 0.043, 0.206 |
| 10% | $8.085{\times}10^{33}$, $8.951{\times}10^{16}$ | 0.097, 0.410 |
| 15% | $8.089{\times}10^{33}$, $8.935{\times}10^{16}$ | 0.139, 0.580 |
| 20% | $6.027{\times}10^{33}$, $8.912{\times}10^{16}$ | 25.39, 0.844 |

| Noise | Coefficients | Error (%) |
|---|---|---|
| 0% | $8.078{\times}10^{33}$, $8.987{\times}10^{16}$ | 0.001, 0.006 |
| 5% | $8.078{\times}10^{33}$, $8.986{\times}10^{16}$ | 0.005, 0.022 |
| 10% | $8.078{\times}10^{33}$, $8.984{\times}10^{16}$ | 0.009, 0.038 |
| 15% | $8.079{\times}10^{33}$, $8.983{\times}10^{16}$ | 0.013, 0.054 |
| 20% | $8.079{\times}10^{33}$, $8.981{\times}10^{16}$ | 0.017, 0.070 |
| 30% | $8.080{\times}10^{33}$, $8.978{\times}10^{16}$ | 0.025, 0.103 |

Finally, LCEN is compared with the feature selection algorithm in ALVEN (Sun & Braatz, 2020), which uses the same basis function expansion, but uses F-tests for feature selection. The "4th-degree, univariate polynomial" dataset is created as per Sun & Braatz (2021), such that $y = X + 0.5X^2 + 0.1X^3 + 0.05X^4 + \epsilon$, 30 $X$ points are available for training, and 1,000 $X$ points are available for testing. These conditions simulate the scarcity of data potentially present in real datasets while ensuring test errors can be predicted with high confidence. Sun & Braatz (2021) created four types of ALVEN models for this prediction. On this same dataset and using the same four types of models, LCEN attained median errors that are typically over 60% smaller than those from ALVEN (Fig. 1). Discussion of these results are included in the Appendix (Section A5.3), including discussions on how LCEN consistently selected the correct *degree* hyperparameter via cross-validation despite the low number of training samples and high noise (Fig. A9).

### 3.2  In empirical datasets, LCEN provides higher predictive performance than many other methods

The applicability of an algorithm to real-world problems is judged only by its performance on real data, as data sparsity or real noise may affect the algorithm's capabilities. In tests performed on empirical data generated by processes with known physical laws, LCEN is observed to display high feature selection performance, consistently selecting the right features with low coefficient relative errors even when the number of features is much higher than the number of samples (Table 2).

The first test of an empirical dataset from a process with a known physical law uses the "CARMENES star data" dataset from Schweitzer et al. (2019). This dataset contains information on temperature ($T$), radius ($R$), and luminosity ($L$) of 293 white dwarf stars. These features are linked together by the Stefan-Boltzmann equation, $L = 4\pi R^2 \sigma T^4$, where $\sigma$ is a constant. Normalizing this equation to values from another star (typically, the Sun), conveniently sets the constant terms to 1. This normalization is applied to the "CARMENES star data" dataset. LCEN with *degrees* from 1 to 10 was applied to this normalized dataset. Despite the very large number of potential features (due to the high *degree* values used), LCEN correctly selected only the $R^2T^4$ feature (Table 2, third column). The coefficient assigned to $R^2T^4$ is 1.0008, which is well within the 2–3% empirical error on these data (as reported by Schweitzer et al. (2019)).

A potential limitation in real datasets is data scarcity. To evaluate the LCEN algorithm in a low-data scenario, the "Kepler's 3rd Law" dataset is created from the original data obtained by Kepler, first published

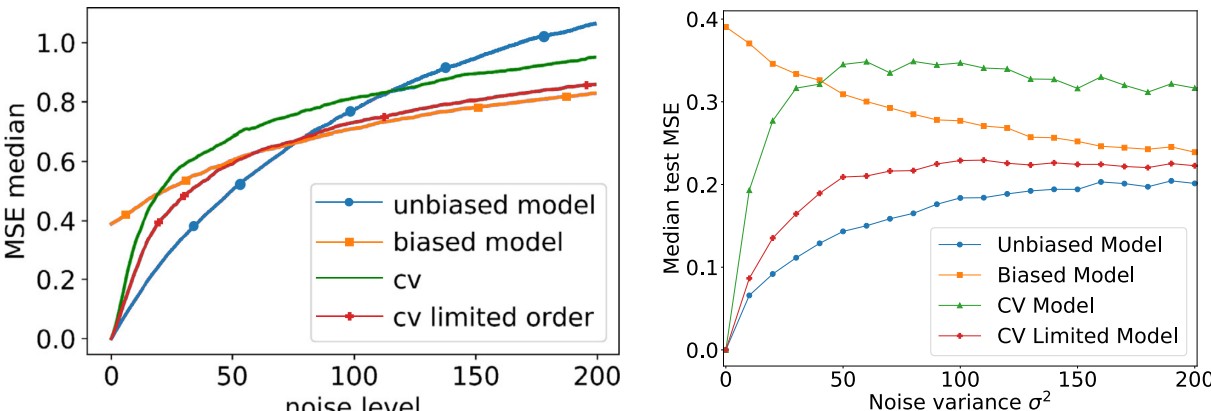

**Figure 1:** Test set median MSE for the "4th-degree, univariate polynomial" dataset. ALVEN results (left, reproduced from Sun & Braatz (2021) with permission) show that the error is monotonically increasing with noise and that the *degree* 4 "unbiased model" is the best at low noise levels, but is displaced by the *degree* 2 "biased model" at higher noise levels. On the other hand, LCEN results (right) show that the median errors converge at higher noises. Furthermore, the LCEN median errors are typically over 60% smaller than the ALVEN median errors, and the *degree* 4 "unbiased model" is always the best model no matter the noise. The "noise level" and "Noise variance $\sigma^2$" terms are equivalent in this figure. Fig. A8 contains interquartile ranges for the LCEN model's test MSEs.

in 1619 and republished in Kepler et al. (1997). From only 6 (slightly inaccurate) measurements, Kepler was able to derive the eponymous Kepler's 3rd law, which states that the period $T$ of a celestial body is related to the semi-major axis of its orbit $a$ by $T = ka^{3/2}$. The constant $k$ depends on the masses of the central and orbiting bodies; however, as the mass of the central body is typically much larger, the mass of the orbiting body is ignored. In this and Kepler's works, $T$ is measured in Earth days, so the constant $k$ is $\approx$365.25 when using modern data and $\approx$365.15 when using Kepler's original data. Again, LCEN with *degrees* from 1 to 10 was used to model this dataset. Despite the low number of data points, LCEN correctly selected only the $a^{3/2}$ feature (Table 2, sixth column). Moreover, the coefficient assigned to that feature was 366.82, an error of only 0.46% relative to Kepler's $k = 365.15$.

As mentioned in Section 2.2, the LCEN, LC, and LCL methods performed much more accurate nonlinear feature selection than LEN or the methods starting with EN (Table 2, third and sixth columns). Moreover, the methods that start with the LASSO are considerably faster, a difference clearly visible with the larger "CARMENES star data" dataset (Table 2, fourth column). Although LC performed perfect feature selection, the coefficients for the true feature it selected were significantly distorted, especially in the "CARMENES star data" dataset. Only LCEN and LCL consistently selected only the correct features with low coefficient errors in these experiments (Table 2, second and third columns).

**Table 2:** Average ($\pm$ standard deviation across 3 CV seeds) results for empirical datasets from processes with known physical laws. True feature %RE refers to the percent relative error of the coefficient of the true features.

| Model | CARMENES star data (293×350) | | | Kepler's 3rd Law (6×130) | | |
|---|---|---|---|---|---|---|
| | True feature %RE | MCC (%) | Runtime (s) | True feature %RE | MCC (%) | Runtime (s) |
| LCEN | 0.08±0.00 | 100±0 | 6.01 | 0.46±0.00 | 100±0 | 3.56 |
| LC | 78.3±0.00 | 100±0 | 5.63 | 3.83±0.31 | 100±0 | 3.12 |
| ENC | 85.3±2.97 | 34.1±2.6 | 55.4 | 72.1±0.38 | 31.3±3.5 | 13.1 |
| LEN | 79.3±2.77 | 16.2±0.4 | 6.87 | 6.59±4.12 | 59.5±6.2 | 3.50 |
| LCL | 0.08±0.00 | 100±0 | 5.73 | 0.46±0.00 | 100±0 | 3.17 |
| ENCEN | 67.5±2.64 | 57.4±0 | 54.6 | 79.1±9.89 | 31.3±3.5 | 13.6 |

The final experiments to compares the performance of LCEN to other algorithms on real datasets from processes with unknown physical laws. As there is no (computational) way to validate the feature selection done by models trained on these datasets, this section focuses on investigating the capabilities of LCEN as a machine learning algorithm by comparing the prediction error and sparsity of different models. Thus, despite the fact that feature selection metrics such as discovery rates or MCCs would be of interest, it is impossible to obtain those metrics for these empirical datasets. Ordinary least squares (OLS), ridge regression (RR) (Tikhonov, 1963), partial least squares (PLS) (Wold, 1975a;b), LASSO, elastic net (EN) (Zou & Hastie, 2005), SCAD (Fan & Li, 2001), MCP (Zhang, 2010), random forest (RF) (Ho, 1995), gradient-boosted decision trees (GBDT) (Friedman, 2001), adaptive boosting (AdaB) (Freund & Schapire, 1997), support vector machine with radial basis functions (SVM) (Boser et al., 1992), fastSparseGAMs (FS-GAMs) (Liu et al., 2022), multilayer perceptron (MLP), MLP with group LASSO (MLP-GL$_1$) (Scardapane et al., 2017), and LassoNet (Lemhadri et al., 2021) were compared with LCEN. To clarify our design choices and the relevance of each individual part of the LCEN algorithm, ablation tests are performed with many of the datasets tested here in Section A4 of the Appendix.

The first dataset analyzed is the "Diesel Freezing Point" dataset (Hutzler & Westbrook, 2000), which is comprised of 395 diesel spectra measured at 401 wavelengths and used to predict the freezing point of these diesels. The dense, nonlinear method SVM had the best prediction performance, with a test RMSE of 4.72°C (Table 3). The next best in performance were LCEN and EN, which had test set RMSEs equal to 4.89°C. The sparsest methods were LCEN, which selected only 37/401 features (9.2%) on average yet had an average prediction error only 3.6% higher than that of the best dense method, LASSO, which performed similarly to LCEN, SCAD, which selected only 22/401 features (5.5%) on average but had an average prediction error 11% higher than that of the best dense method, and FS-GAMs, which selected only 2/401 features (0.5%) but had a prediction error 76.3% higher than that of the best dense method. LCEN, FS-GAMs, and SVM were the only nonlinear methods that had a runtime faster than 10 seconds, comparable to the linear methods PLS and RR (Table 3). LCEN and LASSO are the only methods that simultaneously had a low test RMSE, interpretability, and a fast runtime. However, LASSO has a slightly worse RMSE and less-stable sparsity, as most of its seeds selected more features than those selected in any of the LCEN seeds (Table 3, third column).

**Table 3:** Average (± standard deviation across 3 CV seeds) test results for different models for the "Diesel Freezing Point" dataset.

| Model | Test RMSE (°C) | Features | Runtime (s) |
|---|---|---|---|
| OLS | 11.75 | 401 | 0.09 |
| PLS | 5.21±0.00 | 401±0 | 7.30±0.06 |
| RR | 4.90±0.07 | 401±0 | 6.75±0.02 |
| EN | 4.89±0.06 | 280±209 | 20.99±0.58 |
| LASSO | 4.95±0.10 | 33±10 | 4.52±0.09 |
| SCAD | 5.26±0.11 | 22±7 | 29.3±5.9 |
| MCP | 5.22±0.07 | 30±10 | 33.9±2.8 |
| FS-GAMs | 8.32±0.00 | 2±0 | 3.75 |
| RF | 5.10±0.12 | 393±7 | 471±5 |
| GBDT | 5.21±0.18 | 383±25 | 2,870±32 |
| AdaB | 5.17±0.00 | 304±0 | 23.8±0.5 |
| SVM | 4.72±0.33 | 401±0 | 6.74±0.05 |
| MLP | 4.95±0.14 | 401±0 | 3,839±60 |
| MLP-GL$_1$ | 4.98±0.13 | 401±0 | 10,314 |
| LassoNet | 9.88±0.06 | 401±0 | 47,679 |
| LCEN | 4.89±0.06 | 37±1 | 6.59±0.63 |

We also consider a different scenario: that an end user could prioritize creating very sparse models, even at the expense of increasing the MSE. To simulate this scenario, the LCEN *cutoff* hyperparameter was increased from the value that minimizes the validation MSE to create sparser models. These models have much fewer features, yet their test set RMSEs typically increase by only a small amount (Table A9). This

experiment illustrates how sparsity can be prioritized by the end user to make models with high predictive power while retaining the most critical features.

Finally, we train models on the same dataset but using 10-fold CV instead of 5-fold CV to determine whether there are any performance differences when using more folds. The ratio between test-set RMSEs with 10-fold CV and 5-fold CV equals 1.018±0.059 (Table A10), a value that is indistinguishable from 1, indicating that there is no benefit to using 10-fold CV in this experiment. Furthermore, the use of 10-fold CV increased runtimes by a factor of 1.856±0.282 on average.

LCEN is then tested on the "Concrete Compressive Strength" dataset (Yeh, 1998), which contains the composition and age of 1,030 different types of concrete and their compressive strengths. The relationship between these properties is nonlinear, and past modeling efforts include algebraic expressions and artificial neural networks (specifically, MLPs)[4] (Yeh, 1998; 2006). LCEN's test RMSE is considerably lower than that of the published algebraic models (Table 4, top rows), and nearly the same as for the MLPs (within the noise level). RF has the lowest test RMSE, which is lower by only 9.3%. However, LCEN has the advantage of being interpretable and having the lowest validation RMSE out of all methods tested.

**Table 4:** Average test RMSE ($\pm$ standard deviation across 3 CV seeds) for different models for the "Concrete Compressive Strength" dataset. All machine-learning models selected all 8 features except for FS-GAMs, which selected 4.

| Model | Test RMSE (MPa) |
|---|---|
| Algebraic expression (Yeh, 1998) | 7.79 |
| Linear + interactions model (Yeh, 2006) | 7.43 |
| OLS | 10.26 |
| PLS = RR = EN = LASSO = SCAD = MCP | 10.26±0.00 |
| RF | 5.08±0.02 |
| GBDT | 5.97±0.67 |
| AdaB | 6.95±0.00 |
| SVM | 6.07±0.19 |
| MLP | 5.47±0.08 |
| MLP-GL$_1$ | 5.47±0.09 |
| LassoNet | 16.39±0.17 |
| FS-GAMs | 11.39±0.00 |
| LCEN | 5.55±0.04 |

To assess the performance of LCEN for a dataset caused by human activity instead of physical law, consider the modified "Boston housing" dataset. This dataset contains the median value of owner-occupied houses and many internal and external measurements, such as the per-capita crime rate of the region, the average number of rooms, and the concentration of nitric oxides in the area (Harrison & Rubinfeld, 1978). We modified this dataset to detransform the B variable into its raw value; samples in which this detransformation led to multiple possible values were discarded. In this modified "Boston housing" dataset, the linear models tended to perform very similarly to each other but quite poorly (Table 5). RF and SVM performed relatively well, whereas LassoNet and FS-GAMs had the worst performance. A dense MLP had the lowest test RMSE, but MLP-GL$_1$ was slightly worse. LCEN had a very high performance on this regression task, reaching a test RMSE only 5.5% higher than that of the dense MLP and 2.0% lower than that of the MLP-GL$_1$. LCEN also had the lowest validation RMSE, which was 12% lower than that of the dense MLP. As with the other datasets, LCEN attained higher performance than many other methods while also being completely interpretable.

Finally, the "GEFCom 2014" dataset was used to evaluate the abilities of LCEN in a complex and dynamic task (Hong et al., 2016). Two versions of the "GEFCom 2014" dataset have been published: one that contains

---

[4]No type of validation is mentioned in Yeh (1998), so the test and validation sets are likely the same, making its MLP results overoptimistic. Corroborating this hypothesis, the MLP and MLP-GL$_1$ models trained in this work have higher test-set MSEs.

**Table 5:** Average test RMSE (± standard deviation across 3 CV seeds) for different models for the "Boston housing" dataset.

| Model | Test RMSE (Thousands USD) |
|---|---|
| OLS | 6.38 |
| PLS | 6.50±0.19 |
| RR = EN | 6.42±0.03 |
| LASSO | 6.38±0.00 |
| SCAD | 6.37±0.01 |
| MCP | 6.37±0.01 |
| RF | 5.09±0.13 |
| GBDT | 6.42±0.21 |
| AdaB | 5.67±0.05 |
| SVM | 5.05±0.07 |
| MLP | 4.69±0.07 |
| MLP-GL$_1$ | 5.05±0.32 |
| LassoNet | 9.93±0.02 |
| FS-GAMs | 7.32±0.00 |
| LCEN | 4.95±0.10 |

only energy consumption levels and another that contains the same energy consumption data and also temperature data from multiple weather stations. This work uses the former. "GEFCom 2014" is part of an energy forecasting competition won by a LASSO-like model (Hong et al., 2016). More recently, deep learning has been applied to this problem (Wilms et al., 2018; Gasparin et al., 2022), which produced models with strong 24-hour predictive performance (Gasparin et al., 2022). Despite the strong performance of multiple, complex ANN architectures, LCEN models obtain a 13.1% lower test RMSE on this forecasting task than the state-of-the-art Seq2Seq model from Gasparin et al. (2022) (Table 6, sixth and seventh columns). Unlike the ANNs, LCEN requires only a CPU for training and forecasting, and provides interpretable coefficients. LCEN can also be used for longer forecasts without significant increases in the prediction error, demonstrating a high robustness of the algorithm.

**Table 6:** Test RMSE and relative error for different models for the "GEFCom 2014" dataset. The deep learning models (TCN to Seq2Seq) and their results are from Table 8 of Gasparin et al. (2022). Confidence intervals for LCEN are all ±0 due to LCEN's high resistance to variation due to model initialization changes and the fact that changes in CV seed are not possible when using time series CV. A similar phenomenon occurs with the ARIMA models of Gasparin et al. (2022), and with LASSO/RR/EN/LCEN models in general.

| Model | TCN | RNN | LSTM | GRU | Seq2Seq | LCEN | | | | |
|---|---|---|---|---|---|---|---|---|---|---|
| Hours Forecast | 24 | 24 | 24 | 24 | 24 | 24 | 48 | 72 | 120 | 168 |
| Test RMSE (MW) | 17.2±0.1 | 18.0±0.3 | 19.5±0.5 | 19.0±0.2 | 17.1±0.2 | 14.9 | 18.9 | 21.0 | 23.4 | 24.7 |
| Relative Error (%) | 9.8±0.06 | 10.2±0.2 | 11.1±0.3 | 10.8±0.1 | 9.7±0.1 | 8.5 | 10.7 | 11.9 | 13.2 | 13.9 |

## 4 Discussion

This work introduces LASSO-Clip-EN (LCEN), a nonlinear, interpretable feature selection and machine learning algorithm (Algorithm 1). LCEN was first validated using artificial data (Section 3.1), which provide an initial assessment of the algorithm's performance under multiple, independently controllable conditions. LCEN was then tested with data from processes with known physical laws (Section 3.2) and without known physical laws (Sections 3.2 and A6.1).

Overall, these experiments have demonstrated the applicability of LCEN to a multitude of scientific and nonscientific problems, even those with significant nonlinearities and complexity. On the real data from

processes with known physical laws, LCEN successfully selected only the correct features with very low coefficient errors for all datasets used in this work, effectively rediscovering physical laws solely from data (Table 2). LCEN models were robust to defects in the real data, including noise, multicollinearity, or sample scarcity. LCEN models were typically as accurate as or more accurate than many alternative methods Tables 3–6), yet were also much sparser (Table 3). LCEN models are also easy to interpret, as they display exactly how each input contributes to the final output. This combination of accuracy and interpretability is essential for the deployment of machine-learning models in performance-critical scenarios, from aviation to medicine. Moreover, the additional interpretability can assist in data or model refinement efforts and can make the models robust to changes in data or adversarial input. LCEN is free, open-source, and easy to use, allowing even non-specialists in machine learning to benefit from and use it. Its main limitations are that (1) LCEN is not a universal function approximator, as it can model only the functions present in the expansion of dataset features, (2) its feature expansion algorithm is better suited to numerical data over image or text data, and (3) LCEN is not always as accurate as a dense deep learning method. If enough compute and time are available for model training, users in scenarios that focus on accuracy above anything else or with non-numerical data types may prefer to use a deep learning method.

There are two clear future directions for this work. The first involves evaluating LCEN in classification tasks, as many important problems in science and engineering involve classification. The second involves applying LCEN to automatically generate physical equations for hybrid model architectures (such as physics-constrained or physics-guided ML), which have high potential for scientific applications (Peng et al., 2021; Willard et al., 2022).

## Acknowledgments

This work was partially supported by a Project Award Agreement from the National Institute for Innovation in Manufacturing Biopharmaceuticals (NIIMBL) from the U.S. Department of Commerce, National Institute of Standards and Technology [70NANB17H002, 70NANB21H086]. P.S. was supported by a MathWorks Engineering Fellowship.

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

## A1    Appendix — Feature Expansion Algorithm

---

**Algorithm 2** Feature expansion for LCEN

---

  **Input:** X and y data; hyperparameters *degree, lag, trans_type, interaction, transform_y*
  **if** *lag* $> 0$ **then**
    Append X data from the previous *lag* time steps (samples) to each time step.
    **if** *transform_y* $==$ True **then**
      Append y data from the previous *lag* time steps (samples) to each time step.
    **end if**
    Discard the first *lag* time steps.
  **end if**
  Using sklearn's PolynomialFeatures function, generate polynomial (and interaction if the hyperparameter *interaction* is True) transforms of the X data for the given *degree*.
  **if** *trans_type* $==$ 'all' **then**
    Generate logarithm transforms.
    Generate square root transforms for the features with all values $> 0$.
    Generate inverse $[1/X_k]$ transforms.
    **if** *degree* $\geq 2$ **then**
      Generate transforms with noninteger degrees $[(X_k)^{N+1/2}$ for integers $N$ such that $|N| < $ *degree*$]$ for the features whose every sample is $> 0$.
      Generate the log-inverse transforms $[(\ln X_k)^N/(X_k)^M$ for natural numbers $N$ and $M$ such that $N + M < $ *degree*$]$.
      Generate the log-sqrt-inverse transforms $[(\ln X_k)^N/(X_k)^{M-1/2}$ for natural numbers $N$ and $M$ such that $N + M < $ *degree* $- 1]$ for the features with all values $> 0$.
    **end if**
  **end if**
  **return** the transformed X features

---

## A2    Appendix — Description of datasets used in this work

Three types of data are used in this work: artificial data ["Artificial Linear", "Multicollinear data", "Relativistic energy", and "4th-degree, univariate polynomial"], empirical data from processes with known physical laws ["CARMENES star data" and "Kepler's 3rd Law"], and empirical data from processes with no known physical laws ["Diesel Freezing Point", "Abalone", "Concrete Compressive Strength", "Boston housing", and "GEFCom 2014"]. The artificial data are generated by us as described in the next paragraph. These artificial data are used for an assessment of the feature selection capabilities of the LCEN algorithm and to investigate how properties of the data, such as noise or data range, affect its capabilities. Empirical data from processes with known physical laws are described in Section 3.2 and used to verify whether the LCEN algorithm can rediscover known physical laws using data with real properties. Empirical data from processes with no known physical laws are described in Sections 3.2 and A6.1, and used to compare the machine learning performance of the LCEN algorithm against other linear and nonlinear methods, including deep learning models.

The "Artificial Linear" datasets were created by drawing numbers from a uniform distribution between $-10$ and $10$ in intervals of $0.1$ for the samples $X$ and coefficients $k$ and summing to generate the outputs $y = \sum_{i=1}^{\text{n\_samples}} k_i X_i$. These datasets feature all combinations of $\{100, 500, 1000\}$ samples $\times$ $\{100, 500, 1000\}$ true features $\times$ $\{0\%, 25\%\}$ noise level $\times$ $\{25\%, 50\%, 75\%, 100\%\}$ additional false features. The noise level is defined as mean(added noise/noiseless y)$\times 100\%$. The "Multicollinear data" dataset was created by drawing numbers from a uniform distribution between 1 and 10 to create one variable $X_0$, which was used together with a small amount of noise to create a correlated variable $X_1 = X_0 + \epsilon_1$; finally, they were summed such that $y = 2X_0 + 2X_1 + \epsilon_2$. The "Relativistic energy" dataset was created by drawing numbers from a uniform distribution between 1 and 10 or 1 and 100 for masses, and $5 \times 10^7$ and $2.5 \times 10^8$ for velocities, which represent the energy of a body as $E^2 = c^4 m^2 + c^2 m^2 v^2$. With these velocity numbers, relativistic effects are responsible for 20.4% of the total squared energy on average. The "4th-degree, univariate polynomial" dataset

was created by drawing numbers from a normal distribution with mean 0 and variance 5 and transforming them into the polynomial $y = X + 0.5X^2 + 0.1X^3 + 0.05X^4 + \epsilon$.

All models tested in this work had their hyperparameters selected by 5-fold cross-validation, except for those trained on the "GEFCom 2014" dataset, which used time series cross-validation. The separation between training and testing sets varied depending on the dataset. None of the artificial datasets or datasets containing empirical data from processes with known physical laws have a separate test set, as they are used to investigate the feature selection capabilities of LCEN (which depend only on the training set). For the "Diesel freezing point" dataset, 30% of the dataset was randomly separated to form the test set. For the "Abalone" dataset, the last 1,044 entries (25%) were used as the test set as per Waugh (1995) and Clark et al. (1996). For the "Concrete Compressive Strength" dataset, 25% of the dataset was randomly separated to form the test set as per Yeh (1998). For the "Boston housing" dataset, 20% of the dataset was randomly separated to form the test set. For the "GEFCom 2014" dataset, the data from task 1 were used as the training set and all data from tasks 2–15 were used as the test set.

**Table A1:** Datasets used in this work and their sources. The artificial datasets are used in Section 3.1, the real datasets from processes with known physical laws are used in Section 3.2, and the real datasets from processes with unknown physical laws are used in Section 3.2.

| Dataset Name | Source |
|---|---|
| Artificial Linear | Artificial data generated by us |
| Multicollinear data | Artificial data generated by us |
| Relativistic energy | Artificial data generated by us |
| 4th-degree, univariate polynomial | Artificial data generated by us |
| CARMENES star data | Schweitzer et al. (2019) [link to dataset] |
| Kepler's 3rd Law | Kepler et al. (1997) (Original from 1619) |
| Diesel Freezing Point | Hutzler & Westbrook (2000) [link to dataset] |
| Abalone | Nash et al. (1995) |
| Concrete Compressive Strength | Yeh (1998) [dataset: Yeh (2007)] |
| Boston housing (modified by us) | Harrison & Rubinfeld (1978) [link to dataset] |
| GEFCom 2014 | Hong et al. (2016) [link to dataset] |

## A3 Appendix — List of hyperparameters used in this work

All possible permutations of the hyperparameters below were cross-validated.

1. For the LASSO and ridge regression models: $\alpha = 0$ and 20 log-spaced values between $-4.3$ and 0 (as per `np.logspace(-4.3,0,20)`).

2. For the elastic net (EN) models: $\alpha$ as above and L1 ratios equal to [0, 0.1, 0.2, 0.3, 0.4, 0.5, 0.6, 0.7, 0.8, 0.9, 0.95, 0.97, 0.99].

3. For the SCAD models: $\alpha$ as above and the $a$ parameter (also written as $\gamma$) equal to 3.7, the default value. According to Fan & Li (2001), SCAD is invariant to changes in $a$.

4. For the MCP models: $\alpha$ as above and $\gamma$ equal to [1, 1.5, 2, 2.5, 3, 3.5, 4].

5. For the symbolic regression (SymReg) models: most hyperparameters were set to their default values as per Stephens et al. (2022), except for the following. *population_size* was increased to 2,000, *p_crossover* equal to [0.7, 0.8, 0.9, 0.95], *p_subtree_mutation* equal to [0.01, 0.025, 0.05, 0.1, 0.15], *p_hoist_mutation* equal to [0.01, 0.025, 0.05, 0.1], and *p_point_mutation* equal to [0.01, 0.025, 0.05, 0.1, 0.15] were tested. Because the sum of these probabilities must be $\leq 1$, some combinations are not feasible.

6. For the partial least squares (PLS) models: a number of components equal to all integers between 1 and a limit were used. This limit is either the number of features or 80% of the number of samples, whichever is smaller.

7. For the LCEN models: $\alpha$ and L1 ratios as above. *degree* values equal to [1, 2, 3] were typically used, except when otherwise indicated (such as in the "Relativistic energy" dataset). *lag* = 0 was used, except for the "GEFCom 2014" dataset, which used *lag* = 168. *cutoff* values between $1{\times}10^{-3}$ and $5.5{\times}10^{-1}$ were used; higher values were used only when intentionally creating models with fewer selected features. A *cutoff* = 0 is used in the ablation tests for the LASSO-EN model (Section A4).

8. For the random forest (RF) and gradient-boosted decision tree (GBDT) models: [10, 25, 50, 100, 200, 300] trees, maximum tree depth equal to [2, 3, 5, 10, 15, 20, 40], minimum fraction of samples per leaf equal to [0.01, 0.02, 0.05, 0.1], and minimum fraction of samples per tree equal to [0.1, 0.25, 0.333, 0.5, 0.667, 0.75, 1.0]. For the GBDT models, learning rates equal to [0.01, 0.05, 0.1, 0.2] were also used.

9. For the AdaBoost (AdaB) models: [10, 25, 50, 100, 200, 300] trees/estimators and learning rates equal to [0.01, 0.05, 0.1, 0.2] were used.

10. For the support vector machine (SVM) models: C values equal to [0.01, 0.1, 1, 10, 50, 100], epsilon values equal to [0.01, 0.025, 0.05, 0.075, 0.1, 0.15, 0.2, 0.3], and gamma values equal to [1/50, 1/10, 1/5, 1/2, 1, 2, 5, 10, 50] divided by the number of features in a dataset were used.

11. For the fast sparse GAMs (FS-GAMs): the $L_0 L_2$ penalty, num_gamma = 20, gamma_min = $5{\times}10^{-5}$, gamma_max = 1, and a max_support_size equal to the larger of 20% of the features or 8 were used.

12. For the multilayer perceptron (MLP), MLP with group LASSO (MLP-GL$_1$), and LassoNet models: the hidden layer sizes varied for each dataset. Representing an MLP with one hidden layer as [X], an MLP with two as [X, Y], and an MLP with three as [X, Y, Z], hidden layer sizes of

   - {[800], [400], [200], [100], [800, 800], [800, 400], [800, 200], [800, 100], [400, 400], [400, 200], [400, 100], [200, 200], [200, 100], [100, 100], [800, 800, 800], [800, 800, 400], [800, 800, 200], [800, 800, 100], [800, 400, 400], [800, 400, 200], [800, 400, 100], [800, 200, 200], [800, 200, 100], [400, 400, 400], [400, 400, 200], [400, 400, 100], [400, 200, 200], [400, 200, 100], [200, 200, 200], [200, 200, 100], [200, 100, 100], [100, 100, 100]} were used with the "Diesel Freezing Point" dataset.
   - {[18], [9], [4], [18, 18], [18, 9], [9, 9], [9, 4], [9, 2], [4, 4]} were used with the "Abalone" dataset.
   - {[48], [40], [32], [24], [16], [8], [4], [48, 48], [48, 40], [48, 32], [48, 24], [48, 16], [48, 8], [40, 40], [40, 32], [40, 24], [40, 16], [40, 8], [32, 32], [32, 24], [32, 16], [32, 8], [24, 24], [24, 16], [24, 8], [16, 16], [16, 8], [8, 8], [8, 4], [40, 40, 40], [40, 40, 32], [40, 40, 24], [40, 32, 32], [40, 32, 24], [40, 32, 16], [40, 24, 24], [40, 24, 16], [40, 16, 16], [32, 32, 32], [32, 32, 24], [32, 32, 16], [32, 24, 24], [32, 24, 16], [32, 16, 16], [24, 24, 24], [24, 24, 16], [24, 16, 16], [16, 16, 16], [16, 16, 8], [16, 8, 8], [8, 8, 8]} were used with the "Concrete Compressive Strength" dataset.
   - {[78], [65], [52], [39], [26], [13], [6], [78, 78], [78, 65], [78, 52], [78, 39], [78, 26], [65, 65], [65, 52], [65, 39], [65, 26], [65, 13], [52, 52], [52, 39], [52, 26], [52, 13], [39, 39], [39, 26], [39, 13], [26, 26], [26, 13], [13, 13], [13, 6], [78, 78, 78], [78, 78, 65], [78, 78, 52], [78, 78, 39], [78, 78, 26], [78, 65, 65], [78, 65, 52], [78, 65, 39], [78, 65, 26], [78, 52, 52], [78, 52, 39], [78, 52, 26], [78, 39, 39], [78, 39, 26], [78, 26, 26], [65, 65, 65], [65, 65, 52], [65, 65, 39], [65, 65, 26], [65, 52, 52], [65, 52, 39], [65, 52, 26], [65, 39, 39], [65, 39, 26], [65, 26, 26], [52, 52, 52], [52, 52, 39], [52, 52, 26], [52, 39, 39], [52, 39, 26], [52, 26, 26], [39, 39, 39], [39, 39, 26], [39, 26, 26], [26, 26, 26]} were used with the "Boston housing" dataset.

   Learning rates equal to [0.0005, 0.001, 0.005, 0.01, 0.05], the AdamW optimizer, the ReLU and tanhshrink activation functions, a batch size of 32, weight decay with $\lambda$ equal to [0, 0.01, 0.05, 0.08, 0.1], 100 epochs, and a cosine scheduler with a minimum learning rate equal to 1/16 of the original learning rate with 10 epochs of warm-up were also used. For the MLP-GL$_1$ and LassoNet, regularization parameters equal to [$1{\times}10^{-4}$, $1{\times}10^{-3}$, $1{\times}10^{-2}$] were used.

**Table A2:** Additional features included for each value of the *degree* hyperparameter for a dataset with three features labeled $X_0$, $X_1$, and $X_2$ when the *lag* hyperparameter is set to 0, the *trans_type* hyperparameter is set to 'all', and the *interaction* hyperparameter is set to True. If the *trans_type* hyperparameter were set to 'poly', only the features of the form $(X_k)^n$ and the interaction terms (if *interaction* were still set to True) would be present. A *degree* of $n$ (any natural number) also includes all features from degrees 1 to $n-1$.

| Degree | Sample new features included [for all $k$] | Features after expansion |
|--------|---------------------------------------------|--------------------------|
| 1 | intercept, $X_k, \ln X_k, (X_k)^{1/2}, 1/X_k$ | 13 |
| 2 | $(X_k)^2$, 2-way interactions, $(\ln X_k)^2, (X_k)^{3/2}, \frac{1}{(X_k)^2}, \frac{\ln X_k}{X_k}$ | 37 |
| 3 | $(X_k)^3$, 3-way interactions, $(\ln X_k)^3, (X_k)^{5/2}, \frac{1}{(X_k)^3}, \frac{(\ln X_k)^2}{X_k}, \frac{\ln X_k}{(X_k)^2}$ | 75 |
| 4 | $[\cdots]$ | 129 |
| 5 | $[\cdots]$ | 201 |

## A4 Appendix — Ablation tests

To better clarify the design choices of the LCEN algorithm and highlight the relevance of each individual part of the algorithm, ablation tests are performed. Three ablated algorithms – LASSO-Clip (LC, the thresholded LASSO), EN-Clip (ENC, the thresholded EN), and LASSO-EN (LEN) – are compared with the original LCEN algorithm. Three variant algorithms, LASSO-Clip-LASSO (LCL), EN-Clip-EN (ENCEN), and regular LCEN followed by OLS for debiasing (LCEN→OLS), are also compared. The "Relativistic energy", "Diesel Freezing Point", "Abalone", "Concrete Compressive Strength", and "Boston Housing" datasets are used in the ablation tests in this section; Table 2 provides ablation test results with the "CARMENES star data" and "Kepler's 3rd Law" datasets.

Tests with the "Relativistic energy" dataset show that models with a Clip step had some degree of success with selecting only relevant features (Table A4). However, the ablated algorithms (LC, ENC, and LEN) had much higher prediction errors for the coefficients of the relevant features, even though LC and ENC were able to select only the relevant features. The variant algorithms (LCL, ENCEN, and LCEN→OLS) had performances closer to that of LCEN, but LCL was slightly worse in terms of error. LCEN→OLS was able to estimate the coefficients with a lower error than LCEN, but this improvement in coefficient estimation performance comes with an increase in test-set MSEs in other tasks.

Globally, the models that begin with EN (ENC and ENCEN) are, on average, one order of magnitude slower than LCEN on all datasets (Tables A4–A7). In particular, LCEN is, on average, 10.3-fold faster the thresholded EN (ENC) in our experiments (Table A3). LCEN consistently had the lowest validation RMSE in all datasets, and had the lowest test-set RMSE in all but one dataset. As highlighted by Table A5, LCEN tied for the sparsest and most accurate model out of all ablated and variant algorithms trained with the "Diesel Freezing Point" dataset. Overall, these ablation experiments highlight how LCEN is the optimal algorithm to maximize accuracy and selectivity while maintaining a low runtime.

**Table A3:** Comparison of runtimes for LCEN and ENC, and overall LCEN fold improvement, for the experiments in this work.

| Dataset Name | LCEN (s) | Thresholded EN (ENC) (s) | Fold Improvement |
|---|---|---|---|
| Artificial Linear (1000×1000, 100% additional false features) | 191 | 902 | 4.72 |
| Relativistic energy | 4.79 | 37.1 | 7.75 |
| CARMENES star data | 6.01 | 55.4 | 9.22 |
| Kepler's 3rd Law | 3.56 | 13.1 | 3.68 |
| Diesel Freezing Point | 6.59 | 20.6 | 3.13 |
| Abalone | 19.2 | 297 | 15.47 |
| Concrete Compressive Strength | 39.7 | 800 | 20.15 |
| Boston housing (modified by us) | 167 | 3045 | 18.23 |
| Average fold improvement: 10.29±6.77 | | | |

**Table A4:** Relative error (%) to the ground truth for the "Relativistic energy" dataset with $1 \leq m < 100$ at different noise levels for ablated and variant LCEN algorithms. The first coefficient is for $m^2$ and the second coefficient is for $m^2 v^2$.

| Noise Level | LC | ENC | LEN |
|---|---|---|---|
| 0% | 36.62, 18.08 | 41.52, 20.91 | 43.75, 22.32 |
| 5% | 37.68, 18.85 | 41.30, 21.16 | 43.93, 23.45 |
| 10% | 18.92, 1.647 | 44.31, 23.70 | 1.137, 0.468 |
| 15% | 39.71, 20.61 | 44.31, 23.70 | 46.65, 26.40 |
| 20% | 39.65, 21.13 | 39.99, 21.95 | 45.87, 27.23 |
| 30% | 22.35, 2.603 | 22.93, 2.660 | 7.649, 1.036 |
| Runtime (s) | 3.70 | 37.1 | 5.40 |

| Noise Level | LCL | ENCEN | LCEN→OLS | LCEN |
|---|---|---|---|---|
| 0% | 0.007, 0.012 | 0.001, 0.006 | 0, 0 | 0.001, 0.006 |
| 5% | 0.011, 0.029 | 0.005, 0.022 | 0.004, 0.016 | 0.005, 0.022 |
| 10% | 0.015, 0.045 | 0.009, 0.038 | 0.008, 0.032 | 0.009, 0.038 |
| 15% | 0.019, 0.061 | 0.013, 0.054 | 0.012, 0.048 | 0.013, 0.054 |
| 20% | 0.023, 0.077 | 0.017, 0.070 | 0.016, 0.064 | 0.017, 0.070 |
| 30% | 0.031, 0.109 | 0.025, 0.103 | 0.024, 0.096 | 0.025, 0.103 |
| Runtime (s) | 3.86 | 38.5 | 4.79 | 4.79 |

**Table A5:** Results of different ablated and variant LCEN algorithms for the "Diesel Freezing Point" dataset. Compare with Table 3.

| Algorithm | Test RMSE (°C) | Features | Runtime (s) |
|---|---|---|---|
| LC | 4.86±0.03 | 37.3±0.6 | 4.39 |
| ENC | 7.26±3.62 | 152±119 | 20.6 |
| LEN | 4.92±0.05 | 39.0±0.0 | 6.64±0.64 |
| LCL | 5.03±0.08 | 35.7±3.1 | 4.71±0.08 |
| ENCEN | 5.04±0.22 | 120±86 | 29.2±5.7 |
| LCEN→OLS | 5.02 | 36 | 6.59 |
| LCEN | 4.89±0.06 | 37.0±1.0 | 6.59±0.63 |

**Table A6:** Results of different ablated and variant LCEN algorithms for the "Abalone" dataset. Compare with Table A11.

| Algorithm | Test RMSE (rings) | Features | Runtime (s) |
|-----------|-------------------|----------|-------------|
| LC | 2.1 | 8 | 11.8 |
| ENC | 2.1 | 8 | 297 |
| LEN | 2.1 | 8 | 26.3 |
| LCL | 2.0 | 8 | 12.9 |
| ENCEN | 2.1 | 8 | 308 |
| LCEN | 2.0 | 8 | 19.2 |

**Table A7:** Results of different ablated and variant LCEN algorithms for the "Concrete Compressive Strength" dataset. All models selected all 8 features, but a varying number of transforms of these features. Compare with Table 4.

| Algorithm | Test RMSE (MPa) | Runtime (s) |
|-----------|-----------------|-------------|
| LC | 5.44±0.12 | 24.6 |
| ENC | 8.29±3.60 | 800 |
| LEN | 5.63±0.15 | 44.9 |
| LCL | 5.77±0.31 | 26.4 |
| ENCEN | 5.73±0.15 | 863 |
| LCEN | 5.55±0.04 | 39.7 |

**Table A8:** Results of different ablated and variant LCEN algorithms for the "Boston Housing" dataset. Compare with Table 5.

| Algorithm | Test RMSE (Thousands USD) | Runtime (s) |
|-----------|---------------------------|-------------|
| LC | 5.25±0.07 | 148 |
| ENC | 5.32±0.12 | 3045 |
| LEN | 5.10±0.06 | 167 |
| LCL | 5.31±0.10 | 147 |
| ENCEN | 5.23±0.14 | 3094 |
| LCEN | 4.95±0.10 | 167 |

## A5    Appendix — Additional results with artificial data

### A5.1    "Artificial Linear" datasets

Figures A1–A4 provide plots for the "Artificial Linear" datasets with 0% noise, and Figs. A5–A6 provide plots for the datasets with 25% noise. Other additional results with artificial data follow from Fig. A7 onwards.

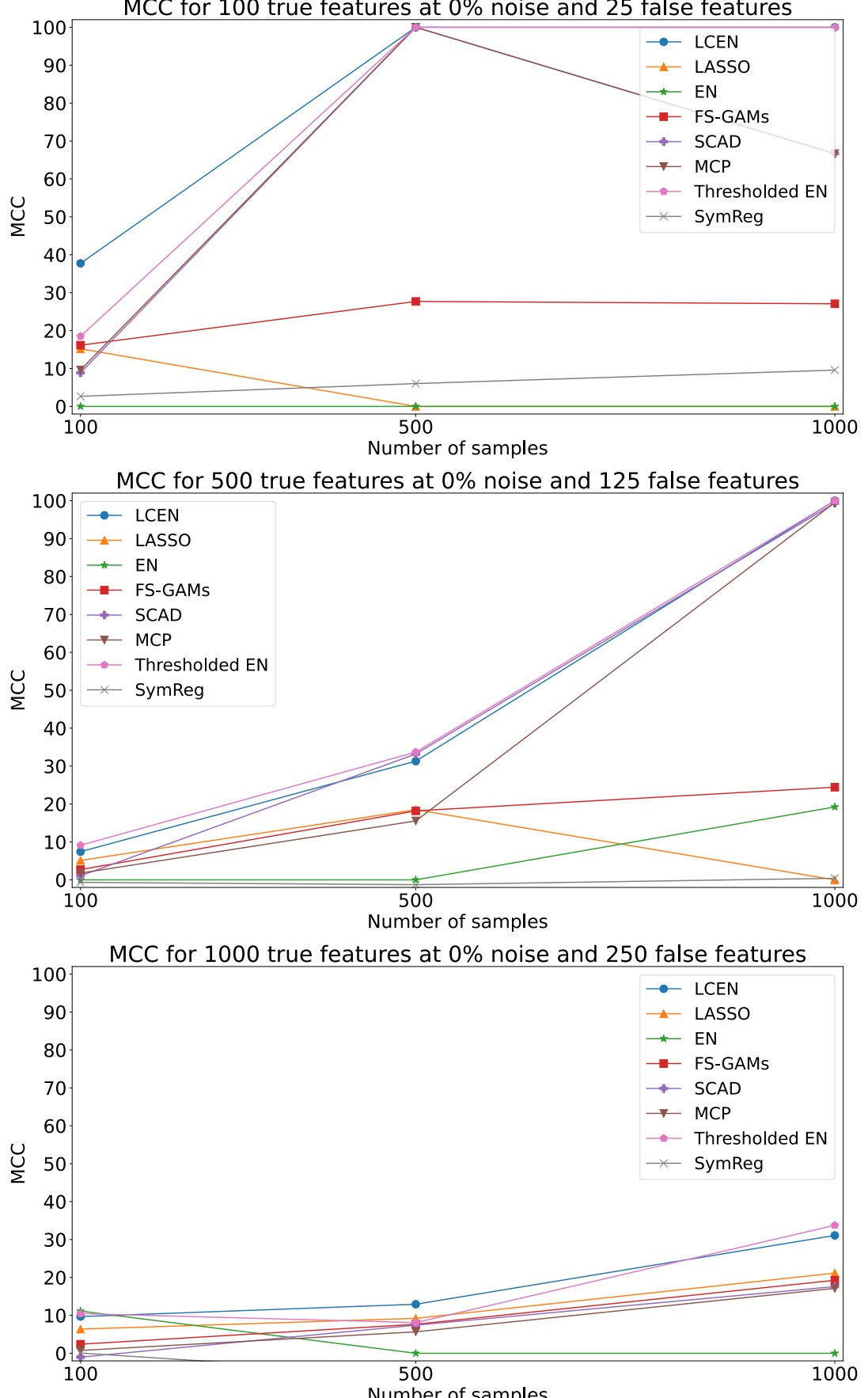

**Figure A1:** Plots of the Matthews Correlation Coefficients (MCCs) for models tested on the "Artificial Linear" dataset with 0% noise and 25% additional false features, as written in each subfigure's title.

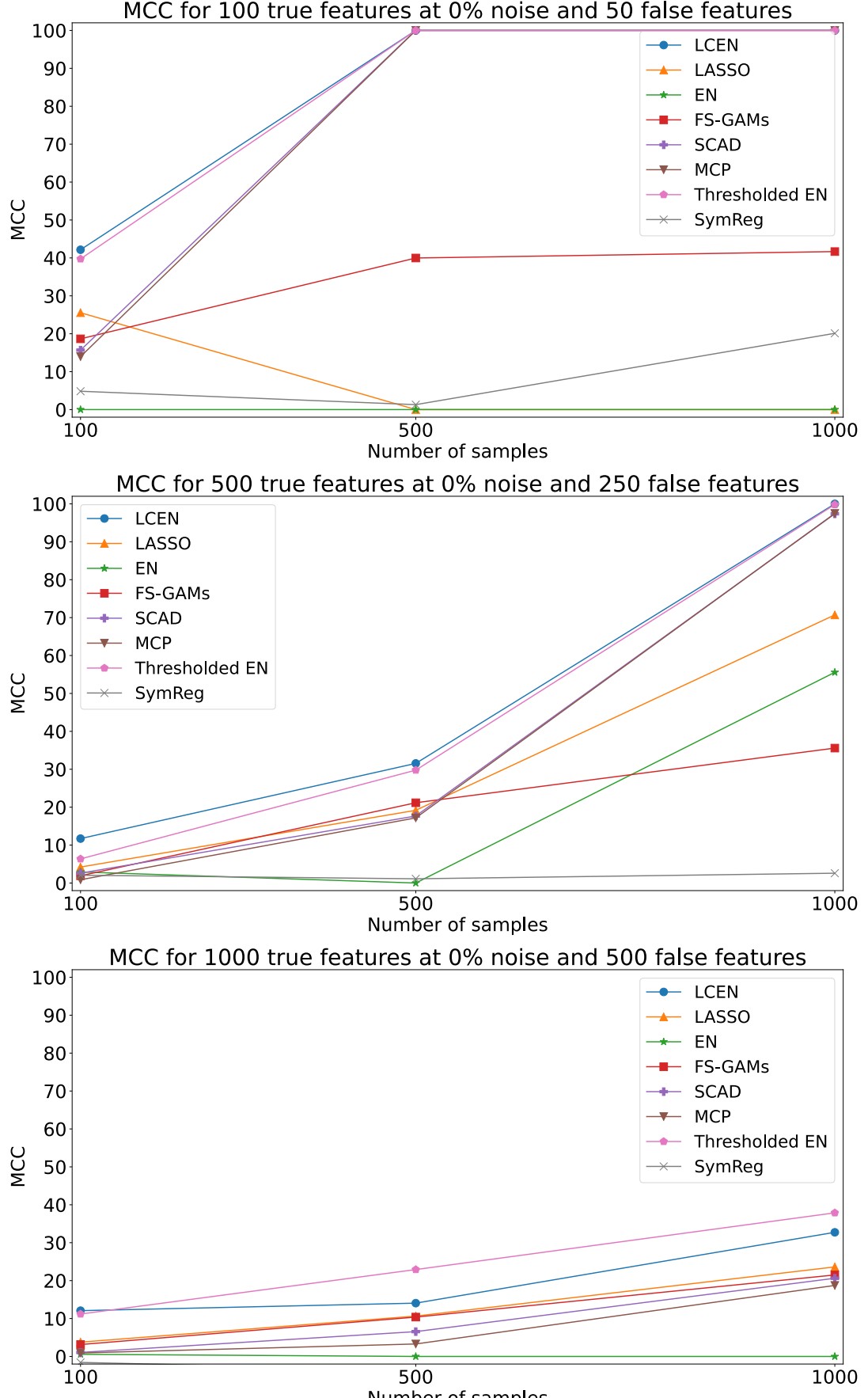

**Figure A2:** Plots of the Matthews Correlation Coefficients (MCCs) for models tested on the "Artificial Linear" dataset with 0% noise and 50% additional false features, as written in each subfigure's title.

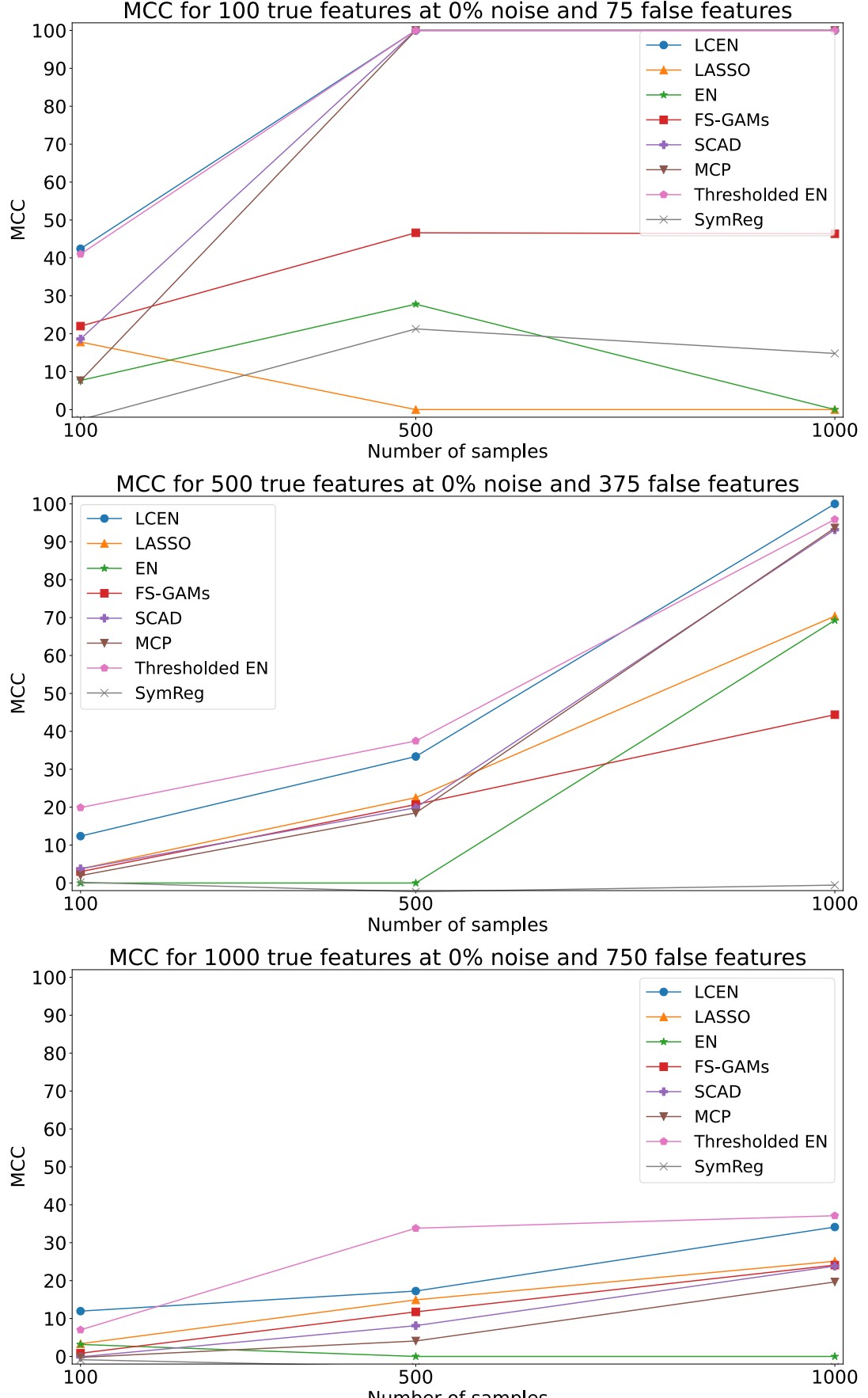

**Figure A3:** Plots of the Matthews Correlation Coefficients (MCCs) for models tested on the "Artificial Linear" dataset with 0% noise and 75% additional false features, as written in each subfigure's title.

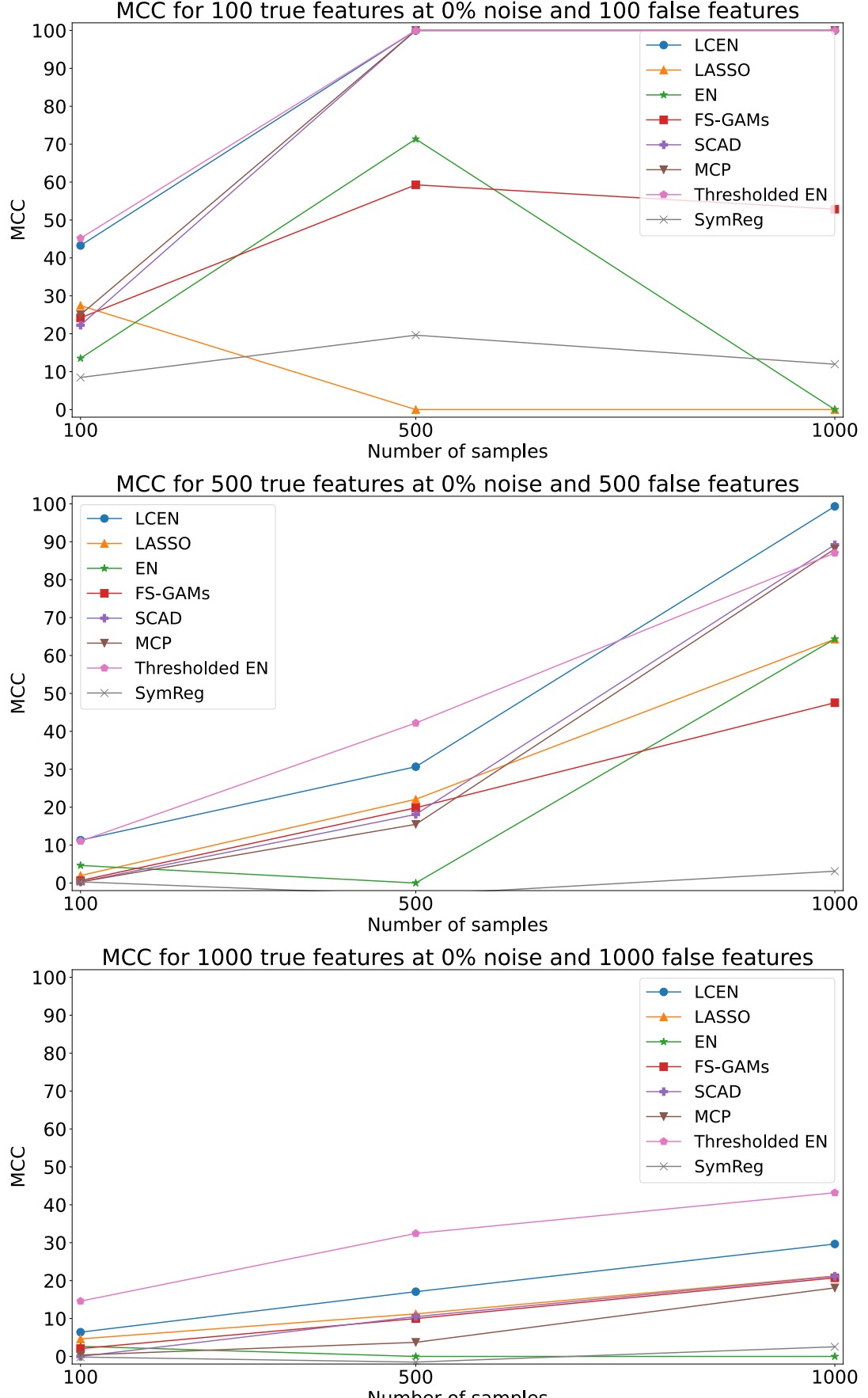

**Figure A4:** Plots of the Matthews Correlation Coefficients (MCCs) for models tested on the "Artificial Linear" dataset with 0% noise and 100% additional false features, as written in each subfigure's title.

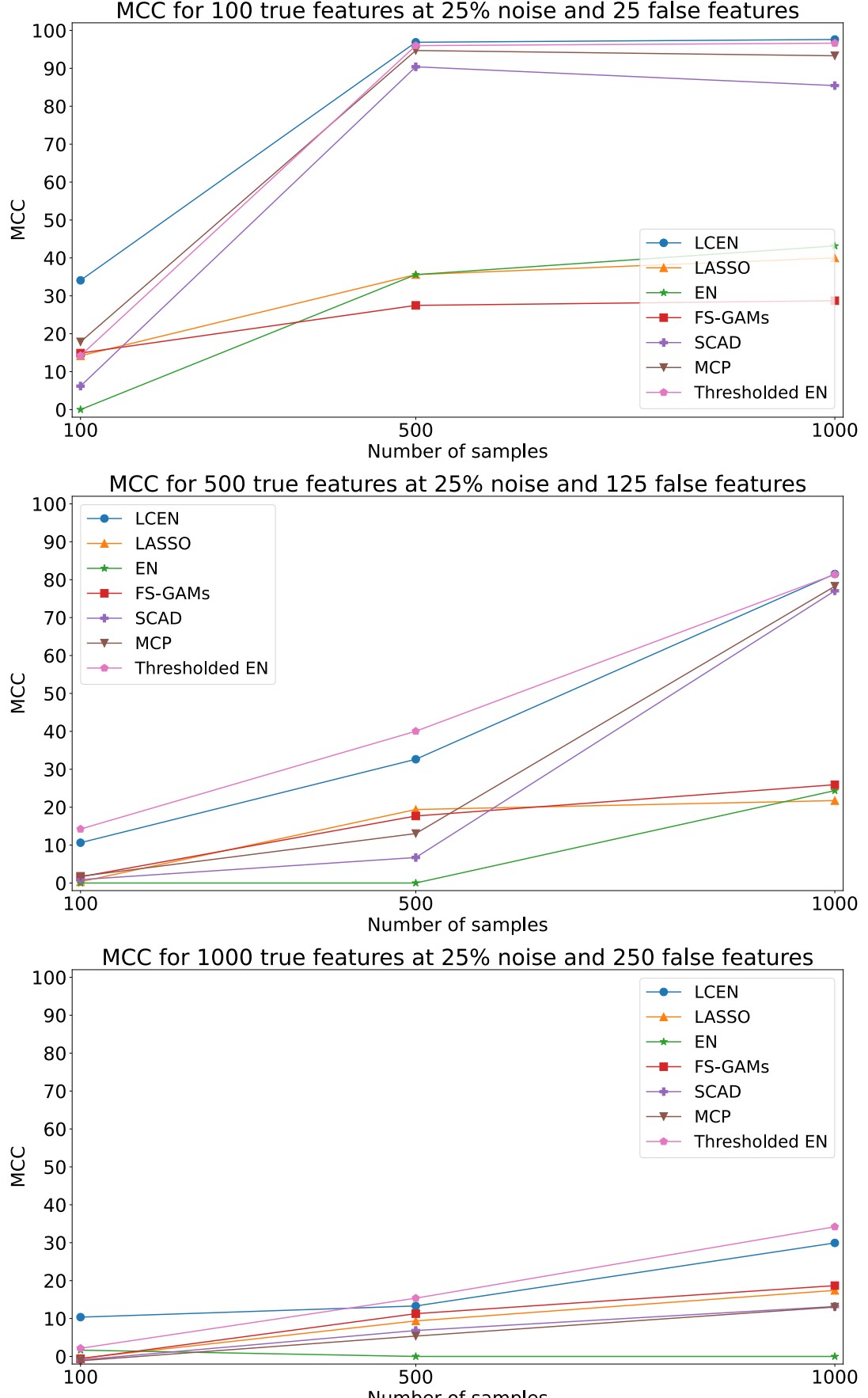

**Figure A5:** Plots of the Matthews Correlation Coefficients (MCCs) for models tested on the "Artificial Linear" dataset with 25% noise and 25% additional false features, as written in each subfigure's title.

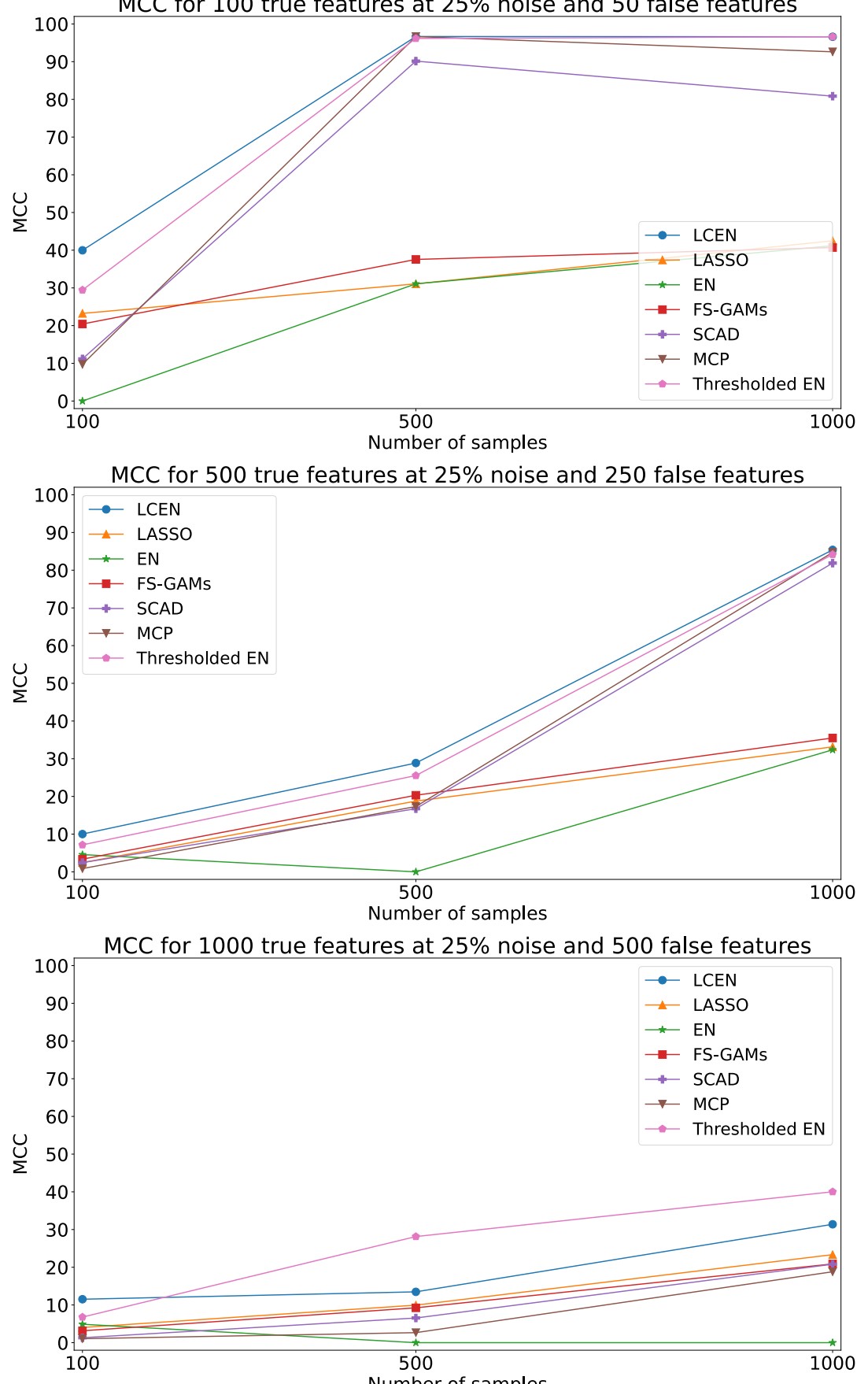

**Figure A6:** Plots of the Matthews Correlation Coefficients (MCCs) for models tested on the "Artificial Linear" dataset with 25% noise and 50% additional false features, as written in each subfigure's title.

## A5.2 "Multicollinear data" dataset

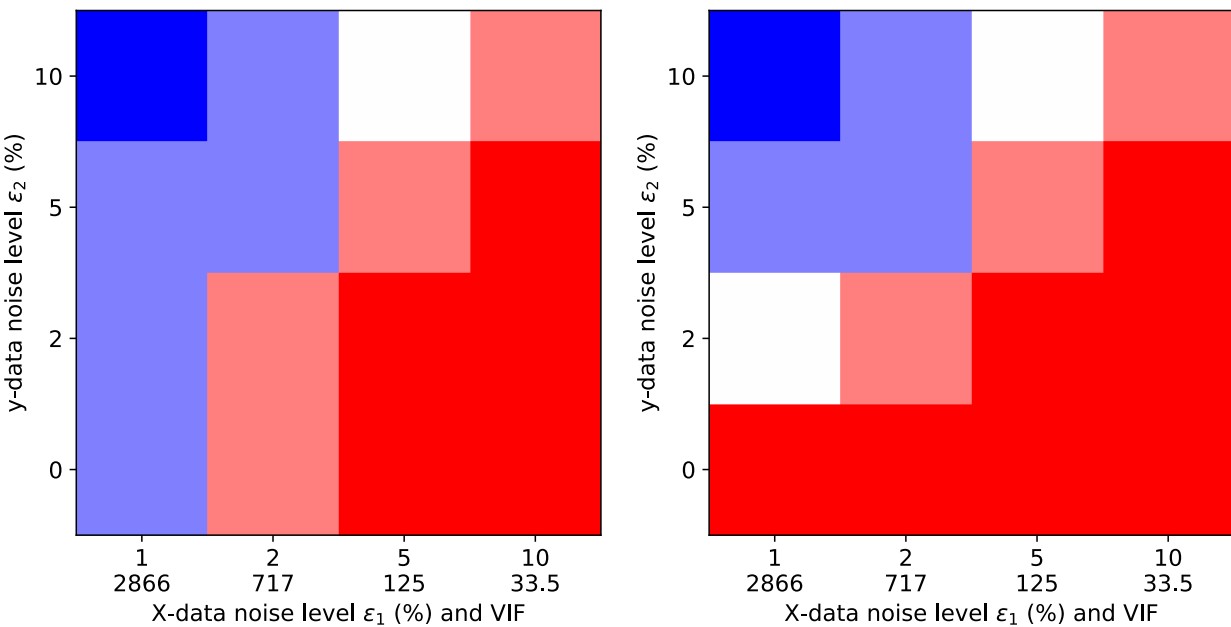

**Figure A7:** LASSO (left) and LCEN/SCAD/MCP (right) model output at different $X$-data noise levels $\epsilon_1$ and $y$-data noise levels $\epsilon_2$ ("Multicollinear data" dataset). Bright red squares indicate both variables were selected and their coefficients had errors $\leq 5\%$. Light red squares indicate that both variables were selected and their coefficients had $5\% <$ errors $\leq 10\%$. White squares indicate that both variables were selected and their coefficients had $10\% <$ errors $\leq 20\%$. Light blue squares indicate that both variables were selected and their coefficients had errors $> 20\%$. Bright blue squares indicate that only one of the variables was selected.

## A5.3 "4th-degree, univariate polynomial" dataset

For the "4th-degree, univariate polynomial" dataset, Sun & Braatz (2020) created four models: one that always uses $degree = 4$ ("unbiased model"), one that always uses $degree = 2$ ("biased model"), one that selects a $degree$ between 1 and 10 based on cross-validation ("cv"), and one that selects a $degree$ equal to 2 or 4 based on cross-validation ("cv limited order"). Sun & Braatz (2021) noted that the $degree$ 4 "unbiased model" was the best at low noise levels, but its error quickly increases, leading to the $degree$ 2 "biased model" becoming the best for noise levels $> 75$ (Fig. 3 of Sun & Braatz (2021); reproduced with permission here as the left subfigure of Fig. 1). The model with $degree$ equal to 2 or 4 "cv limited order" was typically very close in performance to the best model at all noise levels, whereas the model with a $degree$ between 1 and 10 "cv" had lower performance. Sun & Braatz (2021) explain these observations with the bias-variance tradeoff: at low noise levels, models should follow the ground truth as closely as possible; thus, the $degree$ 4 "unbiased model" was the best. However, at sufficiently high noise levels, it becomes impossible to obtain enough signal to compensate for the additional degrees of freedom (variance) in a 4th degree model; thus, the $degree$ 2 "biased model" becomes the best. The $degree$ between 1 and 10 "cv" model had lower performance due to its greater hyperparameter variance, and the $degree$ equal to 2 or 4 "cv limited order" model struck a balance between the "unbiased model" and the "biased model".

Similarly to the models generated using ALVEN, the LCEN model with a $degree$ between 1 and 10 "cv" had the lowest performance and the LCEN model with $degree$ equal to 2 or 4 "cv limited order" had a performance between the $degree$ 4 "unbiased model" and the $degree$ 2 "biased model". However, the $degree$ 4 "unbiased model" was always the best model, no matter the noise level used (Fig. 1). We attribute this considerable reduction in median test MSEs and the superiority of the $degree$ 4 "unbiased model" created by LCEN to the improved feature selection and coefficient estimation algorithm, which is able to better resist

variance due to noise and a large number of hyperparameters. This is corroborated by how the model with a *degree* between 1 and 10 "cv" tended to select *degree* = 4 at lower noise levels and *degree* = 2 at higher noise levels (Figure A9), showing how LCEN can automatically follow the bias-variance tradeoff hypothesis.

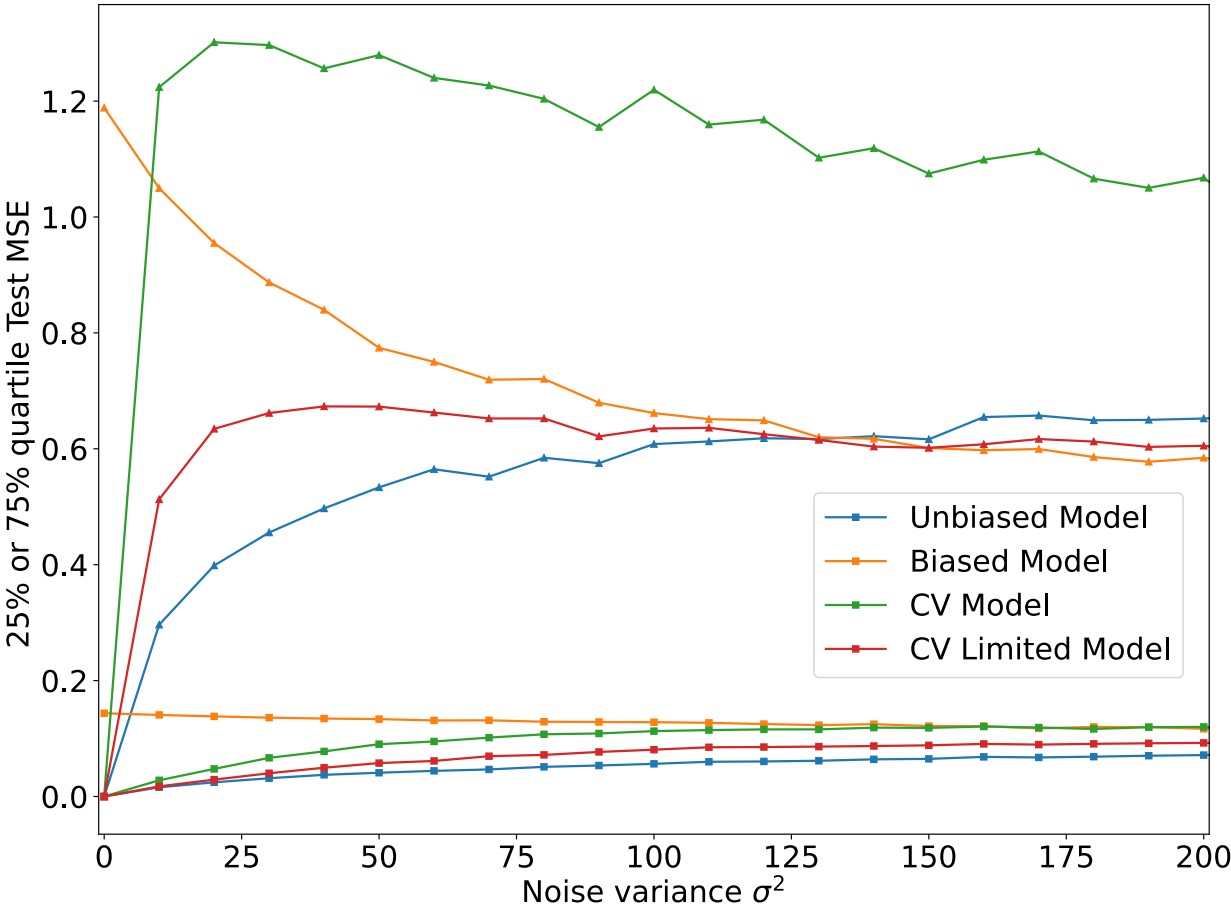

**Figure A8:** 25% (squares) and 75% quartile (triangles) test set MSEs for the LCEN model trained for the "4th-degree, univariate polynomial" dataset. The trends tend to match those from Fig. 1.

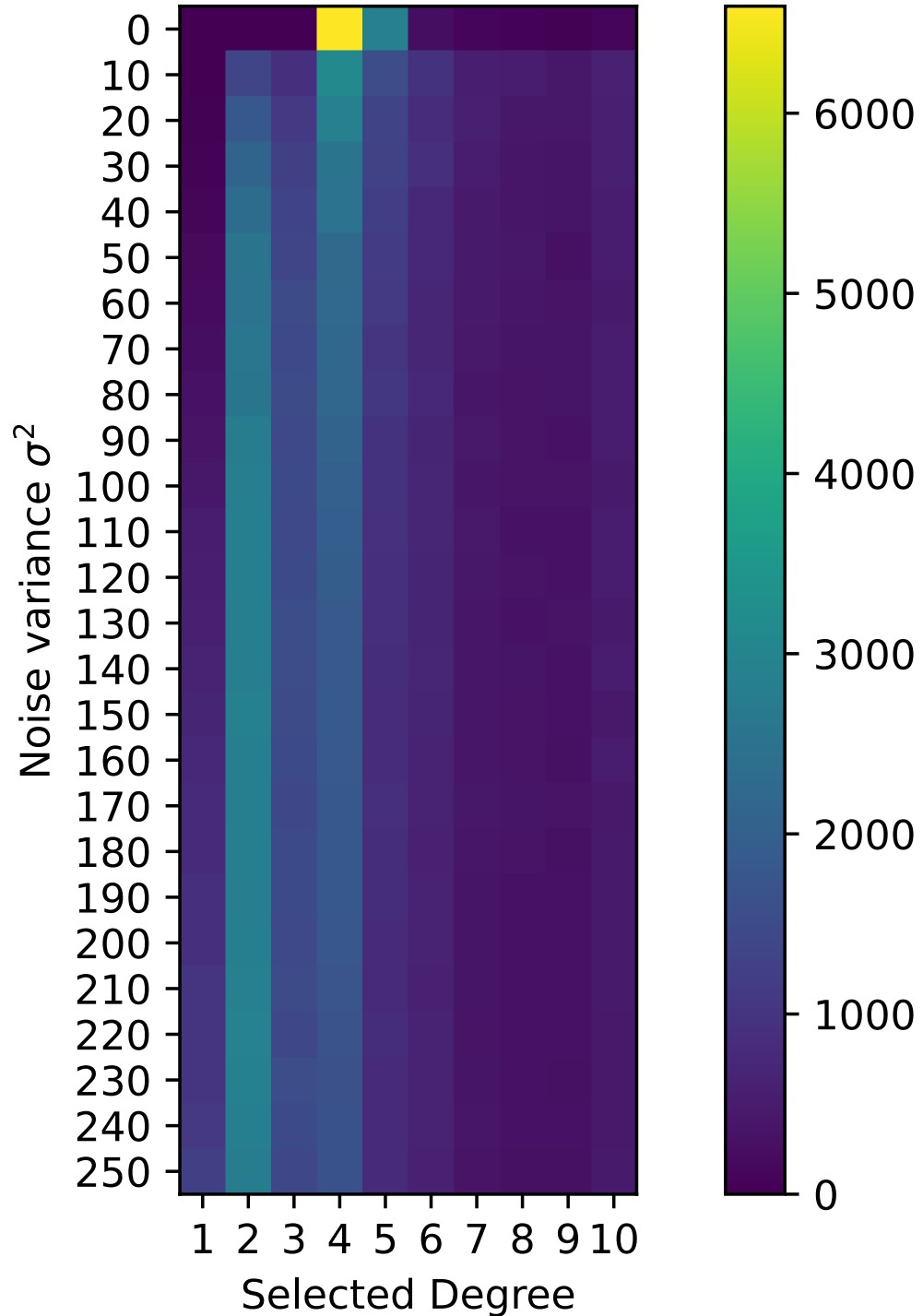

**Figure A9:** *Degrees* selected by the model with a *degree* between 1 and 10 "cv" trained using the LCEN algorithm. At lower noise levels (noise variance $\sigma^2 \leq 30$), LCEN tends to primarily select *degree* = 4. At higher noise levels, there is a shift to primarily select *degree* = 2.

## A6 Appendix — Additional results with empirical data

### A6.1 Datasets for which no physical law is available

The following tables include results for the "Diesel Freezing Point" dataset for two different scenarios: one in which the end-user seeks a sparser model even at the cost of a higher test RMSE (Table A9), and another that compares the results of 5-fold CV with those of 10-fold CV (Table A10). Both scenarios are discussed in the main text (Section 3.2).

**Table A9:** Results of LCEN models with forced higher *cutoff* values trained on the "Diesel Freezing Point" dataset. Compare with Table 3 and the surrounding section.

| Model | Test RMSE (°C) | Features | Runtime (s) |
|---|---|---|---|
| LCEN | 4.89±0.06 | 37±1 | 6.59±0.63 |
| | 4.91 | 29 | 6.27 |
| | 5.52 | 13 | 5.76 |
| | 7.40 | 6 | 5.53 |

**Table A10:** Average (± standard deviation across 3 CV seeds) test results for different models for the "Diesel Freezing Point" dataset. 5-fold CV values are equal to those in Tables 3 and A5.

| Model | Test RMSE (°C) | | Runtime (s) | |
|---|---|---|---|---|
| | 5-fold CV | 10-fold CV | 5-fold CV | 10-fold CV |
| PLS | 5.21±0.00 | 5.10±0.19 | 7.30±0.06 | 17.2±0.1 |
| RR | 4.90±0.07 | 4.87±0.07 | 6.75±0.02 | 11.1±0.03 |
| EN | 4.89±0.06 | 4.88±0.06 | 20.99±0.58 | 43.3±0.3 |
| LASSO | 4.95±0.10 | 4.93±0.05 | 4.52±0.09 | 6.46±0.09 |
| SCAD | 5.26±0.11 | 5.13±0.12 | 29.3±5.9 | 49.3±6.8 |
| MCP | 5.22±0.07 | 5.20±0.04 | 33.9±2.8 | 59.0±3.0 |
| RF | 5.10±0.12 | 5.34±0.13 | 471±5 | 755±8 |
| GBDT | 5.21±0.18 | 6.19±1.51 | 2,870±32 | 6,631±81 |
| AdaB | 5.17±0.00 | 5.17±0.00 | 23.8±0.5 | 49.1±0.4 |
| SVM | 4.72±0.33 | 4.50±0.17 | 6.74±0.05 | 12.4±0.16 |
| MLP | 4.95±0.14 | 5.09±0.05 | 3,839±60 | 7,828±123 |
| LCEN | 4.89±0.06 | 4.88±0.06 | 6.59±0.63 | 11.3±1.2 |
| LEN | 4.92±0.05 | 4.94±0.02 | 6.64±0.64 | 11.3±1.0 |
| LCL | 5.03±0.08 | 5.01±0.08 | 4.71±0.08 | 7.13±0.15 |
| ENCEN | 5.04±0.22 | 5.61±0.58 | 29.2±5.7 | 61.5±9.0 |

Abalone (*Haliotis sp.*) are sea snails whose age can be determined by cutting their shells, staining them, and counting the stained shell rings under a microscope. This process is laborious and error-prone. An alternative is to estimate the number of rings based on readily available physical characteristics, such as weight and size. As before, LCEN was compared with other dense and sparse machine learning models (Table A11). In this problem, OLS, PLS, RR, LASSO, and EN all converged to the OLS solution (that is, no regularization), selecting all 8 linear features and having an RMSE of 2.1 rings. On the other hand, LCEN automatically detected that 2nd degree features would be relevant. The LCEN algorithm model also selected all 8 features and had an RMSE of 2.0 rings, surpassing all linear models and tying with the best nonlinear models in this task. Nonlinear models had test RMSEs between 2.0 and 2.5 rings, but most all lack the interpretability of LCEN.

By increasing the *cutoff* hyperparameter, sparser LCEN models may be generated similarly to what was done in Table A9. An LCEN model with only 3 features had an RMSE of 2.1 rings, and another with only 2 features had an RMSE of 2.2 rings. This experiment further illustrates LCEN's robust feature selection, and how very sparse LCEN models retain significant performance. Furthermore, LCEN models with the same or lower number of selected features had a lower test set RMSE than FS-GAMs.

**Table A11:** Results of different models for the "Abalone" dataset. The number of features that minimizes the cross-validation MSE is 6 for FS-GAMs and 8 for LCEN.

| Model | Test RMSE (rings) | Features |
|---|---|---|
| OLS = PLS = RR = LASSO = EN | 2.1 | 8 |
| SCAD | 2.1 | 8 |
| MCP | 2.1 | 8 |
| SymReg | 2.3 | 3 |
| RF | 2.1 | 8 |
| GBDT | 2.2 | 8 |
| AdaB | 2.3 | 8 |
| SVM | 2.0 | 8 |
| MLP | 2.0 | 8 |
| MLP-GL$_1$ | 2.0 | 8 |
| LassoNet | 2.0 | 8 |
| FS-GAMs | 2.1 | 8 |
| | 2.2 | 6 |
| | 2.4 | 2 |
| LCEN | 2.0 | 8 |
| | 2.1 | 3 |
| | 2.2 | 2 |

## A7    Appendix — Computational resources used

All experiments were done in a personal computer equipped with a 13th Gen Intel® Core™ i5-13600K CPU, 64 GB of DDR4 RAM, and an NVIDIA GeForce RTX 4090 GPU. Runtimes for the models trained on the "Diesel Freezing Point" dataset are provided in Table 3, and runtimes for LCEN and ablated algorithms are provided in the tables of Section A4.

