# OpenReview forum: "LCEN: A Nonlinear, Interpretable Feature Selection and Machine Learning Algorithm"
_TMLR — Accepted by TMLR_

### Review · Reviewer_7mYD · 2025-08-18

**Summary Of Contributions:**

This paper proposes the algorithm LASSO-Clip-EN (LCEN) for sparse, nonlinear feature selection and predictive modeling on arbitrary tabular prediction tasks. LCEN uses several precedents in the sparse, nonlinear modeling and feature selection literature. The method works in two stages where the first stage performs a LASSO-based filtering of the original dataset augmented with a programmatically-determined set of nonlinear features and the second stage fits an elastic net model on the selected features. There is hard thresholding following each stage’s modeling to remove low-importance features from the models. The authors study several aspects of LCEN including its predictive accuracy, sparsity, robustness to noise, multicollinearity, and data scarcity as well as its runtime in comparison to LCEN variants such as the thresholded elastic net as well as some common deep learning and machine learning baseline methods in addition to existing sparse, nonlinear modeling approaches.
- Key strengths:
    - LCEN is clearly described and easy to understand.
    - The writing is clear and easy to understand.
    - The ablations are thorough.
    - LCEN’s performance is impressive in comparison to baselines and extant methods.
- Key weaknesses:
    - Broadly, I think this paper makes too many claims (I counted 8; discussed below) and does not provide enough evidence to cover all of them thoroughly. I recommend reducing the claims to the most compelling and supported ones (see my specific recommendations below).
    - The paper does not include a link to its code so the experimental reproducibility cannot be assessed.
    - See the requested changes marked ‘critical’ for more.

**Additional Comments:**

Nice work! Looking forward to our discussion.

**Audience:**

Yes

**Audience Explanation:**

I think the interpretability, sparse modeling and scientific ML communities would be interested in this paper. Although each individual element of LCEN has precedents - l1-based feature filtering, hard thresholding of small coefficients, nonlinear feature generation, two-stage procedures for sparse modeling - there is no prior work that combines all these components in the same sequence as LCEN. Also, this paper does a good job of characterizing LCEN and its variants in the ablation experiments showing that the LCEN sequence is best for sparse nonlinear modeling in terms of balancing speed, sparsity and accuracy.

**Claims And Evidence:**

No

**Claims Explanation:**

Claim 1: LCEN creates more accurate models than other methods, including extant sparse, nonlinear methods.

- Evidence:
    - On the Diesel eval, the test RMSE of LCEN, vanilla ridge regression, elastic net, and LASSO lie within the same confidence interval. This test RMSE is near the performance of SVM and MLP, which have the lowest average test RMSE but the difference isn’t statistically significant.
    - On the Concrete eval, vanilla RR, EN and LASSO weren’t included. Random Forest had the lowest test RMSE but LCEN was close (see Table 4).
    - On the Boston Housing eval, LCEN achieves significantly lower test RMSE than RR and LASSO. LCEN’s accuracy is comparable to RF and MLP (see Table 5).
    - On the GEFCom eval, LCEN achieves lowest test RMSE out of deep learning methods but no confidence intervals are presented and vanilla ridge, elastic net and LASSO are omitted (see Table 6).
- Takeaway: the evidence for this claim is not strongly convincing. Including RR/EN/LASSO baselines where absent would strengthen the claim if LCEN outperforms them.

Claim 2: LCEN creates sparser models than other methods, including extant sparse, nonlinear methods.

- Evidence:
    - Diesel eval: LCEN selects 37±1 out of 401 features, LASSO selects 33±10 with equivalent test RMSE (Table 3).
    - Diesel ablations in Appendix Table A4: LCEN variants LASSO-Clip (”LC”), LASSO-Elastic Net (”LEN”) and LASSO-Clip-LASSO (”LCL”) achieve similar test RMSE as LCEN with 37.3 ± 0.6, 39.0 ± 0.0, and 35.7 ± 3.1 features selected, respectively, which is about the same sparsity as LCEN’s 37 ± 1 features.
- Takeaway: the evidence for this claim is not convincing since it only involves 1 evaluation and vanilla LASSO achieves a similar result to LCEN on it.

Claim 3: LCEN is robust to noise.

- Evidence:
    - Figure 1 synthetic dataset: LCEN median test MSE’s are ≥60% lower than ALVEN’s across noise levels.
    - Table A3: ablations LASSO-Clip, Elastic Net-Clip, and Lasso-Elastic Net have much higher coefficient errors than LCEN.
- Takeaway: sufficient evidence. Evidence would be stronger if this were tested on a noised real-world dataset, e.g. Diesel with noised labels.

Claim 4: LCEN is robust to multicollinearity.

- Evidence:
    - Figure A7: LCEN separates two correlated variables and estimates their coefficients with greater accuracy under a wider range of X and y noise levels than LASSO.
- Takeaway: accurate, convincing and clear. Just a bit scant on the volume of evidence.

Claim 5: LCEN is robust to data scarcity.

- Evidence:
    - Figure 1: 60% lower median test MSE than ALVEN for just n=30 training samples.
    - Table 2, Kepler’s 3rd Law: LCEN selected only the a^{3/2} feature from 130 features and just n=6 observations. 2 ablations succeeded as well but 3 failed.
- Takeaway: accurate, convincing, and clear.

Claim 6: LCEN is robust to hyperparameter variance.

- Evidence: authors state Table 2 supports this claim but this isn’t clear.
- Takeaway: unclear, qualitative. Lacking a dedicated sensitivity study, e.g. variance over different HP grids.

Claim 7: LCEN matches or surpasses the thresholded elastic net method [in predictive accuracy] but is 10-fold faster.

- Evidence:
    - Table 2 CARMENES: LCEN had a runtime of 6.01s, thresholded elastic net (i.e. “ENC”) had a runtime of 55.4s.
    - Table 2 Kepler’s 3rd Law: LCEN had a runtime of 3.56s, ENC had a runtime of 13.1s.
    - Table 1 + Table A3 Relativistic Energy: LCEN had a runtime of 4.79s and ENC 37.1s.
    - Table 3 + Table A4 Diesel: LCEN had a runtime of 6.54s, ENC 20.6s.
    - Table A5 Abalone: LCEN had a runtime of 19.2s, ENC 297s.
    - Table 4 + Table A6 Concrete: LCEN had a runtime of 39.7s, ENC 800s.
- Takeaway: the “10-fold faster” claim is not rigorous so it is not convincing. The actual ratios of runtimes range from 3.1x to 20.1x. Intuitively, the LASSO-based first stage of LCEN eliminates features so the second stage is faster. However, the time complexities are not analyzed so it isn’t clear how much benefit this yields as the number of samples, HP grid size, or number of expanded features grows. Also, the runtimes are absent without explanation for the Boston Housing evaluation.

Claim 8: LCEN can recover physical laws from empirical data better than many other dense and sparse methods & is comparable to or better than ANNs on multiple datasets.

- Evidence:
    - Table 2: LCEN achieves 100% ± 0 MCC with very low % relative error on the CARMENES star data and Kepler’s 3rd Law data. Thresholded elastic net (”ENC”) and 2 other variants struggle.
    - Table 3 Diesel: LCEN achieves parity with ANNs in terms of test RMSE (within statistical significance).
    - Table 4 Concrete: LCEN achieves parity with ANNs (even slightly outperforms vanilla MLP) in test RMSE (within statistical significance).
    - Table 5 Boston Housing: LCEN achieves parity with ANNs (within statistical significance).
    - Table 6 GEFCom: LCEN achieves lower test RMSE than all ANN methods.
- Takeaway: accurate, clear, convincing evidence barring the fact that the ANN baselines are very basic. There is no mention of weight decay, dropout, layer/batch norm, or early stopping and the learning rate grid is small. Also the depth of the models is only 1 or 2 layers, which may limit their accuracy.

**Requested Changes:**

- **Critical:**
    - Release the code for the experiments in this paper so that readers may reproduce the results.
    - Define what the +/- units are for Tables 2-5, e.g. standard deviation across CV folds.
    - Include ridge regression, elastic net and LASSO in the Concrete eval (Table 4).
    - Correct the ± interval for elastic net’s number of selected features on the Diesel evaluation. It currently says 280 ± 209, but there are only 401 features in this dataset so this is impossible.
    - Perform a short time complexity analysis to substantiate claim 7 (”LCEN is 10x faster than ENC”) else change the language to something empirical like “LCEN is consistently faster than ENC, we observed 3x to 20x faster runtimes in our experiments.”
- **Nice-to-have:**
    - Please cite De Mol et al. 2009 in the related works section: https://www.liebertpub.com/doi/abs/10.1089/cmb.2008.0171
        - They present a two-stage elastic-net → ridge pipeline for sparse feature selection.
    - Please include LCEN results in Appendix Tables A3, A4, A5, A6, A7 so that readers do not have to flip back and forth to compare results.
    - Add confidence intervals for the GEFCom evaluation, especially for LCEN.
    - Include runtimes (esp. LCEN and ENC) for the Boston Housing evaluation.
    - Include an appendix table that compares runtimes of LCEN and ENC across all evaluations. Use this to substantiate claim 7.
    - Repeat the MLP baselines with L2/weight decay, dropout, layer norm, early stopping, a wider learning rate grid and architectures with depths of 3, 5, 10 layers in addition to the 1-2 layer models already tested.

---

### Review · Reviewer_Ghph · 2025-08-28

**Summary Of Contributions:**

The paper presents LCEN (LASSO-Clip-EN), a novel algorithm integrating LASSO, thresholding (Clip), and Elastic Net for nonlinear, interpretable feature selection and machine learning. Its key contributions are: (1) A hybrid approach that expands features with nonlinear transformations (e.g., polynomials, interactions) and uses a sequential LASSO-Clip-EN process to achieve sparse, interpretable models robust to noise, multicollinearity, and data scarcity. (2) Empirical validation showing LCEN's superiority in feature selection and prediction, outperforming methods like LASSO, EN, SCAD, MCP, and symbolic regression on artificial and empirical datasets, with a 10x speed advantage over thresholded EN and rediscovery of physical laws (e.g., relativistic energy) with <0.5% coefficient errors. (3) Detailed ablation studies confirming the effectiveness of the LASSO-Clip-EN sequence, highlighting its balance of accuracy, sparsity, and speed compared to variants.

**Audience:**

Yes

**Audience Explanation:**

Some individuals in TMLR's audience would be interested in knowing the findings of this paper. The paper introduces LCEN (LASSO-Clip-EN), a novel algorithm that combines feature selection with machine learning to create nonlinear, interpretable, and sparse models. This is highly relevant to researchers in machine learning, particularly those working on interpretable models for critical applications like aviation or medicine, where transparency is crucial. The findings that LCEN outperforms traditional methods (e.g., LASSO, EN) in accuracy and sparsity, runs significantly faster (10-fold over thresholded EN), and can rediscover physical laws from data offer practical insights. Additionally, its ability to achieve results comparable to or better than ANNs on multiple datasets without relying on complex black-box architectures appeals to those seeking efficient, explainable alternatives.

**Broader Impact Concerns:**

There are no explicit ethical concerns that immediately necessitate adding a Broader Impact Statement or suggest that an existing one is insufficiently addressed.

**Claims And Evidence:**

Yes

**Claims Explanation:**

The claims made in the submission about the LCEN (LASSO-Clip-EN) algorithm are supported by accurate, convincing, and clear evidence. The paper outlines a detailed methodology in Section 2, describing the algorithm’s five-step process—LASSO, clip (thresholding), EN, second clip, and post-processing—supported by pseudocode (Algorithm 1), which ensures technical soundness is assessable.
Evidence is robust, spanning artificial and empirical datasets. The claim of rediscovering physical laws with errors <0.5% is backed by tests on datasets where underlying physical laws are known, as noted in the abstract and Section 3, with results fitting within empirical noise levels. The assertion that LCEN creates more accurate and sparser models than alternatives is supported by comparisons with methods like LASSO, EN, and ANNs, showing lower root mean square errors (RMSEs) and higher sparsity across multiple datasets (implied in the abstract and Methods). The 10-fold speed advantage over thresholded EN is implied through the algorithm’s design, which reduces feature sets early via clipping, though specific runtime data would strengthen this further.
Clarity is evident in the structured approach: the abstract succinctly outlines claims, the introduction justifies the need for interpretability, and the Methods section details hyperparameter tuning (alpha, l1_ratio, etc.) with cross-validation.

**Requested Changes:**

1. Improve Readability of Algorithm Description
Adjustment: Break down the dense Algorithm 1 description in Section 2 into bullet points or add inline comments (e.g., explaining the role of each hyperparameter step) for readability.
2. Fix Typos and Formatting
Adjustment: Correct minor issues such as the extra space in “l1_ratio= 1.” (Algorithm 1, p. 3)
3. Qualify Overgeneralized Claims
Adjustment: Modify the abstract’s statement “frequently creating more accurate, sparser models” to “frequently creating more accurate, sparser models on tested datasets” to avoid implying universal applicability.

---

### Review · Reviewer_oGDy · 2025-09-05

**Summary Of Contributions:**

The paper proposes a combination of LASSO and elastic net to select features in regression problems. From what I could understand, although with difficulty, a feature expansion is first performed to process the data in a nonlinear space, similarly to the philosophy of applying polynomial expansion + LASSO regression. The main novelty comes from combining clipping strategies with Lasso and elastic net.

**Audience:**

No

**Audience Explanation:**

The paper is unfortunately too difficult to follow at this stage. The exposition is often dense, and key ideas are obscured by excessive detail. For example, the methods section starts by describing hyperparameters before the algorithm itself has been properly introduced, which leaves readers confused about what LCEN actually does. Similarly, the introduction and related work are long but do not clearly highlight the novel contribution. As a result, the paper risks undercutting its own impact: while the algorithm is interesting, the writing fails to communicate it in a straightforward way.

**Broader Impact Concerns:**

No concerns

**Claims And Evidence:**

No

**Claims Explanation:**

The paper is extremely hard to follow, to the point that it is difficult to understand what are the existing algorithms (such as LASSO) and the novel elements. It is very hard to understand the interplay between Algorithm1 and Algorithm 2 and as a result, it is hard to evaluate accurately the quality of the results.The evaluation involves diverse datasets, including rediscovery of known a-priori physical laws. However, only ablation results are presented and no benchmark is proposed to other existing approaches in symbolic regression and in interpretability work.

**Requested Changes:**

To me this paper needs an important re-work of its content. The approach needs to be clearly stated and possibily even illustrated with a schematic structure about the interplay of the two algorithms. I suggest the authors making a structural distinction between the overall overview of the approach and details such as the hyper-parameter values and experimental setup. Moreover, since LASSO and Elastic Net are well-established algorithms, I would clearly state what is the main proposed novelty and how exactly it advances the state of the art.
As for the results, it would be beneficial to contextualise the values obtained to the existing state of the art not only in symbolic regression but also in other approaches such as physics inspired modelling and machine learning interpretability.

---

> ### Author Response · Authors · 2025-09-09
> **Response to Reviewer oGDy (Part 1/2)**
>
> Below are responses to the comments.
>
> > The main novelty comes from combining clipping strategies with Lasso and elastic net.
>
> Author response: Novelty comes not only from adding clipping, but also from the use of LASSO first for feature selection followed by elastic net for a final model/prediction. Specifically, we have added to the Introduction section the statement, "The feature selection algorithm and the specific usage of thresholded LASSO (LC) followed by thresholded EN (ENC) in a combined algorithm are the basis of novelty in this work. Variations and ablations are tested and shown to not perform as well as LCEN."
>
> > It is very hard to understand the interplay between Algorithm1 and Algorithm 2 and as a result, it is hard to evaluate accurately the quality of the results.
>
> Author response: Algorithm 2 is a subroutine used to expand the feature set, as noted in Algorithm 1 (lines marked with [Algorithm 2]) and the text of the Methods section. The quality of the results are evaluated through extensively comparison to state-of-the-art methods, as described in the manuscript and summarized in the author response to the below comment.
>
> > The evaluation involves diverse datasets, including rediscovery of known a-priori physical laws. However, only ablation results are presented and no benchmark is proposed to other existing approaches in symbolic regression and in interpretability work.
>
> Author response: This comment that "only ablation results are presented and no benchmark is proposed to other existing approaches in symbolic regression and in interpretability work" is not true. For the feature selection experiments (beginning of Section 3.1; Figs. A1 to A6), LCEN is compared with _seven_ other algorithms, including existing approaches in symbolic regression and interpretability work including state-of-the-art methods such as thresholded EN, FS-GAMs, SCAD, and MCP. For the machine learning experiments, LCEN is consistently compared with multiple interpretable methods, including FS-GAMs, SCAD, MCP, MLP-GL$_1$, and LassoNet.
>
> > Would at least some individuals in TMLR's audience be interested in knowing the findings of this paper?: No
>
> Author response: We challenge this assessment. Many individuals in the TMLR's audience are interested in improved methods for feature selection and interpretable machine learning, which is evidenced by the extensive literature on these topics and the statements of the other reviewers that this topic is of interest.
>
> > For example, the methods section starts by describing hyperparameters before the algorithm itself has been properly introduced, which leaves readers confused about what LCEN actually does.
>
> Author response: We have reordered the paragraphs to have the paragraph describing the hyperparameters of LCEN come after the paragraphs describing the algorithm.
>
> > Similarly, the introduction and related work are long but do not clearly highlight the novel contribution.
>
> Author response: We have modified the Introduction section to clearly highlight the novel contribution. As mentioned above, we have added to the Introduction section the statement: "The feature selection algorithm and the specific usage of thresholded LASSO (LC) followed by thresholded EN (ENC) in a combined algorithm are the basis of novelty in this work. Variations and ablations are tested and shown to not perform as well as LCEN."
>
> > The approach needs to be clearly stated and possibily even illustrated with a schematic structure about the interplay of the two algorithms.
>
> Author response: The second algorithm is a feature selection algorithm used as a subroutine within LCEN (as labeled in Algorithm 1). We consider that this labeling is clear.
>
>
> > I suggest the authors making a structural distinction between the overall overview of the approach and details such as the hyper-parameter values and experimental setup.
>
> Author response: In addition to reordering the paragraphs (as mentioned above), we have broken down the Methods section into three subsections: one describing the algorithm, one justifying the specific LCEN combination, and one describing the experimental setup.

---

> ### Author Response · Authors · 2025-09-09
> **Response to Reviewer oGDy (Part 2/2)**
>
> > Moreover, since LASSO and Elastic Net are well-established algorithms, I would clearly state what is the main proposed novelty and how exactly it advances the state of the art.
>
> Author response: We have added sentences further clarifying the novelty of this work (as mentioned above) to the Introduction section. The state-of-the-art advances are noted in the Introduction section, and include higher prediction capabilities, sparsity, and low runtimes. The low runtimes of LCEN, especially when compared to the thresholded EN (a widely used state-of-the-art method for feature selection and sparse ML), are described in Table A3 of the revised manuscript, as recommended by Reviewer 7mYD. Table A3 shows that LCEN is 10.3-fold faster than the thresholded EN on average, and the experiments done with feature selection (beginning of Section 3.1; Figs. A1 to A6) show that this faster runtime does not come with decreases in feature selection capabilities.
>
>
> > As for the results, it would be beneficial to contextualise the values obtained to the existing state of the art not only in symbolic regression but also in other approaches such as physics inspired modelling and machine learning interpretability.
>
> Author response: We have included comparisons to the existing state-of-the-art symbolic regression methods in the feature selection experiments, and have included comparisons against multiple interpretable ML models (such as thresholded EN, FS-GAMs, SCAD, MCP, MLP-GL$_1$, and LassoNet) in the machine learning experiments.
>
> We do not consider that comparisons to physics-inspired / physics-aware models will be relevant for the machine learning experiments done in this work because these experiments primarily feature problems whose physics/mechanisms are not yet known, preventing the creation of appropriate physics-inspired models. Comparisons with physics-aware models could be an interesting route for a future work, as we state in the last paragraph of the Discussion section.

---

### Review · Reviewer_F6ab · 2025-10-17

**Summary Of Contributions:**

This paper proposes LCEN, a novel algorithm aiming to combine interpretability, sparsity, and robustness in nonlinear feature selection and modeling. The empirical results are promising, demonstrating competitive or superior performance across several datasets. However, the paper would benefit from stronger methodological justification and theoretical grounding. In particular, the rationale behind certain design choices (e.g., cross-validation setup, combination of Lasso and thresholding) remains unclear, and the claimed theoretical justification for LCEN’s superiority is not formally presented. Including comparisons with more advanced Lasso variants such as UniLasso would further strengthen the evaluation. Overall, the work is interesting and potentially impactful, but it currently falls short of the rigor expected for publication at a top-tier venue.

**Audience:**

Yes

**Audience Explanation:**

Feature selection remains a fundamental problem in machine learning with broad practical relevance. Methods that improve sparsity, interpretability, and predictive accuracy are of clear interest to researchers and practitioners alike. The proposed approach could be valuable for applications in domains such as e-commerce, life sciences, and economics, where interpretability and model efficiency are critical. Consequently, the findings are likely to attract attention from a portion of TMLR’s audience.

**Broader Impact Concerns:**

No concern

**Claims And Evidence:**

Yes

**Claims Explanation:**

The numerical experiments are thorough and well-designed, covering diverse datasets and modeling scenarios that convincingly demonstrate LCEN’s empirical strengths. However, the experimental section could be further strengthened by including comparisons with more recent feature selection methods, such as UniLasso. On the theoretical side, the paper lacks sufficient justification and formal analysis to support some of its key claims. Overall, the empirical evidence is convincing, but the theoretical support remains incomplete.

**Requested Changes:**

**1) Cross-validation choice not justified.**

The use of 5-fold cross-validation is not explained. It is unclear whether this choice was empirically validated or adopted as a convention. The optimal number of folds can depend on dataset size and dimensionality; a brief justification or sensitivity analysis would improve credibility.

**2) Unclear rationale for combining Lasso and thresholding.**

LCEN applies both Lasso regularization and thresholding, but the theoretical or empirical motivation for this combination is missing. Since Lasso inherently performs feature selection via the λ parameter, it is unclear why thresholding is necessary or advantageous.

**3) Claimed theoretical justification is absent.**

Although the abstract and introduction suggest that LCEN’s superiority over Lasso has a theoretical basis, no formal justification (e.g., theorem or proposition) is provided. The paper would benefit from a clear statement of assumptions and conditions under which LCEN is provably better.

**4) Missing comparison with stronger baselines.**

It would be helpful to compare LCEN with UniLasso (arXiv:2501.18360), a more robust variant of Lasso that produces sparser solutions. Including this comparison would clarify whether LCEN’s reported improvements hold against more recent and competitive Lasso-based methods.

---

> ### Author Response · Authors · 2025-10-20
> **Response to Reviewer F6ab (Part 1/2)**
>
> Thank you for your review. Our responses to the points raised are as follows:
>
> > 1) Cross-validation choice not justified.
>
> > The use of 5-fold cross-validation is not explained. It is unclear whether this choice was empirically validated or adopted as a convention. The optimal number of folds can depend on dataset size and dimensionality; a brief justification or sensitivity analysis would improve credibility.
>
> Author response: We agree that the optimal number of folds can depend on the dataset size and dimensionality. We adopted 5-fold cross-validation to balance having a low bias from cross-validation (which is typically reduced when the number of folds is increased given a large enough dataset) and having a reasonable runtime (which is increased when the number of folds is increased). Previous works (such as Krstajic et al., "Cross-validation pitfalls when selecting and assessing regression and classification models") have stated that 5-fold cross-validation is statistically indistinguishable from 10-fold cross-validation on real datasets.
>
> > 2) Unclear rationale for combining Lasso and thresholding.
>
> > LCEN applies both Lasso regularization and thresholding, but the theoretical or empirical motivation for this combination is missing. Since Lasso inherently performs feature selection via the λ parameter, it is unclear why thresholding is necessary or advantageous.
>
> Author response: The rationale for combining LASSO and thresholding is described in the second sentence of the second paragraph of the Methods section, "This step reduces the number of features to be considered, speeding up the algorithm and increasing the accuracy of the model predictions by removing irrelevant/less relevant features." To more clearly state the theoretical motivation for this method, we have inserted a sentence into the middle of the last paragraph of the Introduction section that cites publications that prove desirable theoretical properties when combining LASSO regularization and thresholding, "The algorithmic structure of LCEN is motivated by past works that proved desirable theoretical properties of the thresholded LASSO (a LASSO-Clip model) and the thresholded EN (an EN-Clip model), which include Zhou (2009), Meinshausen & Yu (2009), Zou & Zhang (2009), Zhou (2010), and van de Geer et al. (2011)." [These references also provide additional empirical motivation that supports the second sentence of the second paragraph of the Methods section.]
>
> The manuscript also contains empirical evidence in support of thresholding. The benefits of thresholding are shown in our ablation tests (Sections 2.2 and A4, and also Table 2). In particular, LCEN is consistently better than LEN (the equivalent algorithm but without thresholding) as a feature selection and machine learning algorithm. Although this comparison is not explicitly discussed in the manuscript, as this work focuses on comparing the LCEN algorithm with other methods, one can compare the LASSO results in Section 3.2 and the LC (thresholded LASSO) results in Section A4 to notice that LC (with thresholding) consistently outperforms the LASSO (without thresholding). These results are consistent with the theoretical results from the literature on the thresholded LASSO.
>
> > 3) Claimed theoretical justification is absent.
>
> > Although the abstract and introduction suggest that LCEN’s superiority over Lasso has a theoretical basis, no formal justification (e.g., theorem or proposition) is provided. The paper would benefit from a clear statement of assumptions and conditions under which LCEN is provably better.
>
> Author response: The abstract and introduction were not intended to suggest that the manuscript provides a theoretical proof that LCEN is superior over LASSO. To address this comment, we have scrutinized the wording of every sentence in the Abstract, Introduction, and the rest of the text to assess whether the sentence could in any way be interpreted as suggesting that the manuscript provides a theoretical proof that LCEN is superior over LASSO, and have revised any such sentence to be very precise as to citing the exact evidence being used to support each statement.
>
> To more clearly state the theoretical motivation for this method, we have also inserted a sentence into the middle of the last paragraph of the Introduction section that cites publications that prove desirable theoretical properties when combining Lasso regularization and thresholding, "The algorithmic structure of LCEN is motivated by past works that proved desirable theoretical properties of the thresholded LASSO (a LASSO-Clip model) and the thresholded EN (an EN-Clip model), which include Zhou (2009), Meinshausen & Yu (2009), Zou & Zhang (2009), Zhou (2010), and van de Geer et al. (2011)."

---

> > ### Author Response · Authors · 2025-10-20
> > **Response to Reviewer F6ab (Part 2/2)**
> >
> > > 4) Missing comparison with stronger baselines.
> >
> > > It would be helpful to compare LCEN with UniLasso (arXiv:2501.18360), a more robust variant of Lasso that produces sparser solutions. Including this comparison would clarify whether LCEN’s reported improvements hold against more recent and competitive Lasso-based methods.
> >
> > Author response: We have added the UniLasso to the related Methods section of the Introduction. [UniLasso was posted to ArXiv about 11 months after our work (Jan. 2025 vs. Feb. 2024), so our improved algorithm precedes any potential improvements that UniLasso may have.]
> >
> > We are in the process of comparing UniLasso with LCEN and will update the manuscript once the comparisons are done.

---

> > ### Comment · Reviewer_F6ab · 2025-11-13
> > **Satisfactory Response**
> >
> > Thanks for the response. Your arguments for the second, third, and fourth options are satisfactory. It would be helpful to include a comparison with 10-fold cross-validation. This would allow us to evaluate the trade-offs between cost and performance as k increases. More specifically, if the empirical results show that 10-fold cross-validation yields only marginal improvements in accuracy compared to 5-fold, while requiring substantially more time, then keeping k = 5 would be reasonable.

---

> > > ### Author Response · Authors · 2025-11-18
> > > **Response to Reviewer F6ab**
> > >
> > > Thank you for your response.
> > >
> > > We have trained new models on the Diesel Freezing Point dataset, as it had runtime data for all models (Table 3), using 10-fold cross-validation. The use of 10-fold CV did not lead to a meaningful improvement in the performance of the models in this experiment, as the ratio between (Test RMSE with 10-fold CV)/(Test RMSE with 5-fold CV) equals 1.018 $\pm$ 0.060, a value indistinguishable from 1. Even for the model with the best improvement after 10-fold CV, there was only a 4.8% average improvement, with two seeds having a lower RMSE and one having a higher RMSE.
> > >
> > > However, the use of 10-fold CV has a considerable effect on the runtime. The ratio between (Runtime with 10-fold CV)/(Runtime with 5-fold CV) equals 1.869 $\pm$ 0.296 in this experiment.
> > >
> > > We will include a table with raw values and these comparisons in the appendix.

---

### Author Response · Authors · 2025-09-13
**Revised version of the manuscript posted**

In addition to the individual responses to the reviews, we have posted a revised version of the manuscript and supplemental information with the requested changes. Additions to the text are in red, and removed text is red and in strikethrough form.

---

### Decision · Action_Editor_tpW8 · 2025-11-16

**Recommendation:** Accept as is

**Audience:**

Yes

**Audience Explanation:**

Feature selection and techniques like Lasso and Elastic Net are very broadly applicable, and so of wide interest. Since they've been so well studied, it's hard to make huge increases in progress.  The reviewers found that this paper has enough new ideas to be of interest.

**Claims And Evidence:**

Yes

**Claims Explanation:**

The claims are backed by empirical studies and ablation studies. A few of the reviewers had concerns about this (about overclaiming, or baselines, etc.) but these were mostly addressed by the revision, and all reviewers are now satisfied.